# Antigen presentation plays positive roles in the regenerative response to cardiac injury in zebrafish

João Cardeira-da-Silva [1,2,3] ✉, Qianchen Wang[1,2,3], Pooja Sagvekar [1,2], Janita Mintcheva[4,5], Stephan Latting[1], Stefan Günther [2,3,6], Radhan Ramadass[1], Michail Yekelchyk[3,6], Jens Preussner[3,7], Mario Looso [2,3,7], Jan Philipp Junker [4,8,9] & Didier Y. R. Stainier [1,2,3] ✉

In contrast to adult mammals, adult zebrafish can fully regenerate injured cardiac tissue, and this regeneration process requires an adequate and tightly controlled immune response. However, which components of the immune response are required during regeneration is unclear. Here, we report positive roles for the antigen presentation-adaptive immunity axis during zebrafish cardiac regeneration. We find that following the initial innate immune response, activated endocardial cells (EdCs), as well as immune cells, start expressing antigen presentation genes. We also observe that T helper cells, a.k.a. Cd4+ T cells, lie in close physical proximity to these antigen-presenting EdCs. We targeted Major Histocompatibility Complex (MHC) class II antigen presentation by generating *cd74a; cd74b* mutants, which display a defective immune response. In these mutants, Cd4+ T cells and activated EdCs fail to efficiently populate the injured tissue and EdC proliferation is significantly decreased. *cd74a; cd74b* mutants exhibit additional defects in cardiac regeneration including reduced cardiomyocyte dedifferentiation and proliferation. Notably, *Cd74* also becomes activated in neonatal mouse EdCs following cardiac injury. Altogether, these findings point to positive roles for antigen presentation during cardiac regeneration, potentially involving interactions between activated EdCs, classical antigen-presenting cells, and Cd4+ T cells.

Ischemic heart disease, including myocardial infarction (MI), remains a major cause of death worldwide[1]. In non-fatal scenarios, the lack of blood perfusion and consequent loss of cardiomyocytes together with myocardial remodeling cause extensive damage and scarring which leads to cardiac insufficiency[2]. This scar tissue, combined with the inability of cardiomyocytes to divide, hinders the formation of new and functional myocardium. An adverse immune response, which takes place during myocardial remodeling, contributes to the further damage of the infarcted and non-infarcted tissues[3]. It has been well documented that severe and persistent inflammation during ischemia

[1]Department of Developmental Genetics, Max Planck Institute for Heart and Lung Research, Bad Nauheim, Germany. [2]DZHK German Centre for Cardiovascular Research, Partner Site Rhine-Main, Bad Nauheim, Germany. [3]Cardio-Pulmonary Institute (CPI), Bad Nauheim, Germany. [4]Max Delbrück Center for Molecular Medicine in the Helmholtz Association, Berlin Institute for Medical Systems Biology, Berlin, Germany. [5]Humboldt University of Berlin, Berlin, Germany. [6]Bioinformatics and Deep Sequencing Platform, Max Planck Institute for Heart and Lung Research, Bad Nauheim, Germany. [7]Bioinformatics Core Unit (BCU), Max Planck Institute for Heart and Lung Research, Bad Nauheim, Germany. [8]DZHK German Centre for Cardiovascular Research, Partner Site Berlin, Berlin, Germany. [9]Charité – Universitätsmedizin Berlin, Berlin, Germany. ✉e-mail: joao.cardeira-da-silva@mpi-bn.mpg.de; didier.stainier@mpi-bn.mpg.de

and reperfusion results in detrimental cellular events, including increased cardiomyocyte apoptosis[4]. Over the past decade, considerable attention has also been given to the adaptive immune response to myocardial ischemia, with some publications proposing it as a promising target to limit cardiac damage[5,6]. However, the role of T cells, specifically T helper (T_h) cells, a.k.a. CD4[+] T cells, following MI is controversial[7–10]. Upon MI, autoreactive CD4[+] T cells are activated, leading to pathological outcomes[8,9]. Paradoxically, CD4[+] T cell-deficient mice exhibit impaired healing and reduced survival following MI[10]. Classically, CD4[+] T cell activation is facilitated by antigen-presenting cells (APCs) including B cells, dendritic cells, and monocytes/macrophages, which present antigens through Major Histocompatibility Complex (MHC) class II molecules[7,11,12]. APCs have been previously identified in zebrafish and shown to activate T cells, thereby indicating a conserved process of antigen presentation[13]. However, the antigen presentation mechanisms, particularly those involving MHC class II, that trigger detrimental or cardioprotective adaptive immune programs following cardiac damage are not fully understood.

Increasing effort has been put into studying regenerative organisms towards identifying immunomodulation strategies to boost cardiac regeneration[14]. As opposed to adult mammals, adult zebrafish are able to regenerate ventricular tissue following several types of injury[15–18]. Upon cryoinjury, a regenerative program involving epicardial[19] and endocardial[20] cell (EdC) activation, coronary revascularization[21,22], cardiomyocyte dedifferentiation and proliferation[16,19,23], and progressive resolution of transient collagen accumulation[16,24], takes place to form new and functional heart tissue. Similar to what happens after MI in mammals[25], the zebrafish immune response to cardiac injury involves the early recruitment of neutrophils, followed by macrophages and T cells[26,27]. Yet, it is well established that the type and timing of such a response play major roles in defining the regenerative outcomes post-injury[14,27–35]. While the immune system has been the focus of multiple studies in the context of cardiac regeneration[27,30,32,34,35], particular immune aspects driving the regenerative responses remain relatively unexplored, especially those related to the antigen presentation-adaptive immunity axis. Antigen presentation involves several intracellular processes, some of which take place in APCs and require the activity of key molecules such as Cd74. Although it has been implicated in several adaptive immune processes, Cd74 plays a vital role in the assembly and trafficking of MHC class II molecules during antigen presentation[36].

In zebrafish, regulatory T cells (T_regs) have been shown to promote cardiac regeneration by stimulating cardiomyocyte proliferation[37]. In addition, a comparative transcriptomic study revealed the increased expression of genes related to lymphocyte activation in the regenerating zebrafish heart, but not in the medaka heart, which has a lower regenerative capacity[30]. Furthermore, immune preconditioning in zebrafish leads to an adaptive pro-regenerative response of the cryoinjured heart[38], which may involve the adaptive immune system. Altogether, harnessing antigen presentation, and consequently the adaptive immune response, may hold promise in promoting tissue regeneration in non-regenerative scenarios.

Here, we report the activation of MHC class II antigen presentation genes in both immune cells and EdCs in the regenerating zebrafish heart. Interestingly, we found that during this process, Cd4[+] T cells populate the injured tissue and are in close physical proximity to activated EdCs. Blocking MHC class II antigen presentation by inactivating both cd74a and cd74b resulted in decreased infiltration of Cd4[+] T cells in the central part of the injured tissue. This phenotype correlated with impaired occupancy of the injured tissue by the endocardium as well as reduced EdC proliferation. Moreover, we find that cd74a; cd74b mutants display impaired cardiomyocyte dedifferentiation and proliferation. Overall, our data reveal the activation of antigen presentation genes in the endocardium after cardiac injury and point to positive roles for the antigen presentation-adaptive immunity axis

during zebrafish cardiac regeneration. Notably, a similar activation of Cd74 expression in EdCs is observed in neonatal mice after cardiac injury.

## Results

### Antigen presentation gene upregulation following cardiac cryoinjury

To analyze the overall immune response during zebrafish cardiac regeneration, we FACS-sorted Tg(mhc2dab:EGFP)[+] and Tg(ptprc:DsRed)[+] cells (i.e., single and double positive; Supplementary Fig. 1) from i) uninjured ventricles, ii) ventricles at various time points post-cryoinjury [i.e., 6, 24, and 72 hours post-cryoinjury (hpci) as well 7, 14, and 30 days post-cryoinjury (dpci)], and iii) a 72 hours post-sham sample, and performed scRNA-seq (Supplementary Fig. 2a, b; Supplementary Table 1). Collectively, these two transgenes label the bulk of the immune cells[39,40], as shown in uninjured and 24 hpci ventricles (Supplementary Fig. 2c, d). With this strategy, we generated a small-sized dataset that allowed the identification of the major immune cell groups (Supplementary Figs. 2b, 3a, b and 4a, b). Interestingly, while we observed the presence of APCs, as expected due to their expression of the mhc2dab:EGFP transgene, we also found several other clusters displaying a strong MHC Class II antigen presentation signature including the expression of major histocompatibility complex class II integral membrane alpha chain gene (mhc2a), and CD74 molecule, major histocompatibility complex, class II invariant chain a (cd74a) and b (cd74b) (Supplementary Figs. 3b and 4c, d; Supplementary Data 1). These gene expression data point to a role for antigen presentation during zebrafish heart regeneration.

### Activated endocardial cells also express MHC class II antigen presentation genes during cardiac regeneration

During antigen presentation, APCs stimulate the adaptive immune response by activating naïve Cd4[+] T cells through the presentation of antigens via MHC class II molecules[11] (Fig. 1a). To investigate antigen presentation in the regenerating zebrafish heart, we first analyzed the temporal expression profile of antigen presentation genes (i.e., mhc2a, cd74a and cd74b) in the scRNA-seq dataset of sorted immune cells (Supplementary Fig. 5a). We observed increased expression of these genes starting at 72 hpci, following a transient decrease (Supplementary Fig. 5a). We then generated an antigen presentation reporter line, Tg(cd74a:Gal4ff), to spatially resolve APCs. This reporter labeled cells mostly in the injured tissue, and its expression was particularly enriched at 120 hpci and 7 dpci (Supplementary Fig. 5b). Importantly, we found that at 120 hpci, in addition to macrophages (Fig. 1b), cd74a:Gal4ff was also expressed by activated EdCs, as shown by co-immunostaining with the activated EdC marker Aldh1a2[20] (Fig. 1c; Supplementary Fig. 5c). The MHC class II antigen presentation transgene mhc2dab:EGFP, which is expressed in immune cells[39], was also found to be expressed in activated EdCs in cryoinjured hearts, especially from 120 hpci onwards, although at variable levels (Fig. 1d). mhc2dab:EGFP expression was mostly observed in the injured tissue, and a few scattered mhc2dab:EGFP[+] cells, likely immune cells, were observed in the remote tissue. In uninjured ventricles, only a few activated (i.e., Aldh1a2[+]) EdCs were observed and these activated EdCs were also mhc2dab:EGFP[+] (Fig. 1d). Next, we FACS-sorted EdCs using the Et(krt4:EGFP) line[41] (Supplementary Fig. 6) and used them for RT-qPCR analysis of the antigen presentation genes mhc2a, cd74a, and cd74b (Fig. 1e). We observed basal levels of expression of these genes in EdCs from uninjured hearts (Fig. 1f–h), altogether suggesting that antigen presentation may be active in some EdCs even in homeostatic conditions. Following cryoinjury, we initially (i.e., at 24 hpci) observed decreased expression of all three genes compared with uninjured ventricles, followed by an increase starting at least as early as 120 hpci (Fig. 1f–h). The mRNA levels of these genes were further increased at 14 dpci, suggesting a potential and sustained role for EdCs in antigen

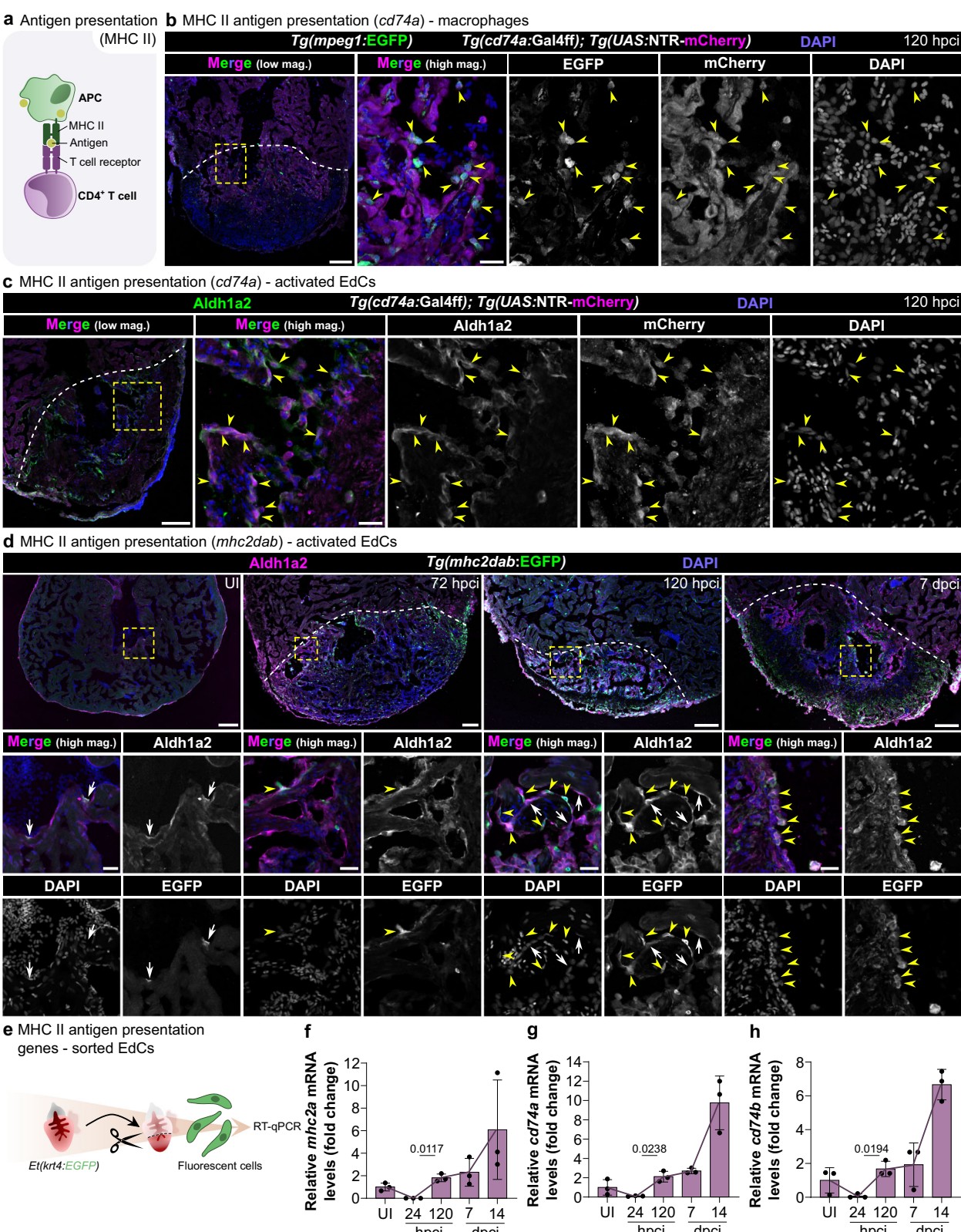

**a** Antigen presentation (MHC II)

**b** MHC II antigen presentation (*cd74a*) - macrophages

**c** MHC II antigen presentation (*cd74a*) - activated EdCs

**d** MHC II antigen presentation (*mhc2dab*) - activated EdCs

**e** MHC II antigen presentation genes - sorted EdCs

presentation at later stages, possibly when some classical APCs (e.g., macrophages) have been depleted from the injured tissue[27]. RT-qPCR analysis from whole injured ventricles also revealed decreased mRNA levels of *cd74a* and *cd74b* at 24 hpci compared with uninjured ventricles, followed by a steady increase as assessed up to 14 dpci (Supplementary Fig. 7a–c), consistent with the data on sorted EdCs

(Fig. 1f–h) and with the scRNA-seq data of immune cells, which further show high expression of *mhc2a*, *cd74a*, and *cd74b* at 30 dpci (Supplementary Fig. 5a). We also took advantage of a published dataset comparing the transcriptomes of the regenerating zebrafish heart and the medaka heart, which has a lower regenerative capacity[30]. We found that while *mhc2a*, *cd74a* and *cd74b* expression was upregulated in the

**Fig. 1 | Antigen presentation genes are activated in leukocytes and endocardial cells during zebrafish cardiac regeneration. a** Schematic illustration of the classical model of MHC class II antigen presentation and the activation of CD4⁺ T cells. **b** Confocal images of a cryosectioned ventricle from an adult *Tg(cd74a:Gal4ff); Tg(UAS:NTR-mCherry); Tg(mpeg1:EGFP)* zebrafish at 120 hpci, immunostained for mCherry and EGFP, showing *cd74a*:mCherry expression in macrophages (yellow arrowheads); two independent experiments with similar results. **c** Confocal images of a cryosectioned ventricle from an adult *Tg(cd74a:Gal4ff); Tg(UAS:NTR-mCherry)* zebrafish at 120 hpci, immunostained for mCherry and Aldh1a2, showing *cd74a*:mCherry expression also in activated EdCs (yellow arrowheads); two independent experiments with similar results. **d** Confocal images of cryosectioned ventricles from *Tg(mhc2dab:EGFP)* uninjured adult zebrafish and at various time points after cryoinjury, immunostained for EGFP and Aldh1a2, showing transgene

expression in activated EdCs (yellow arrowheads, EGFP^high; white arrows, EGFP^low), especially starting at 120 hpci; two independent experiments with similar results. **e–h** Experimental design (**e**) and graphs showing the relative mRNA levels of the antigen presentation genes *mhc2a* (**f**), *cd74a* (**g**), and *cd74b* (**h**) in the absence of injury and at various time points after injury in sorted *Et(krt4:EGFP)*⁺ EdCs, revealing an initial decrease in expression upon cryoinjury followed by an increase starting at 120 hpci. Dots in the graphs represent individual ventricles, and the bars represent the mean ± SD; *n* = 3 biologically independent samples for all time points; two-tailed Welch's *t* test between 72 and 120 hpci (*P* values included in the graphs); Ct values included in Supplementary Table 6. Yellow dashed rectangles and squares outline the magnified areas; dashed lines mark the border of the injured tissue. Scale bars: 100 μm (low magnification); 20 μm (high magnification). UI uninjured, APC antigen-presenting cell, MHC II major histocompatibility complex class II.

cryoinjured zebrafish heart, this upregulation, at least for *cd74a* and *cd74b*, did not occur to the same extent in the cryoinjured medaka heart (Supplementary Fig. 7d–f), suggesting a potential role for Cd74 during cardiac regeneration. We then analyzed a neonatal mouse heart scRNA-seq dataset[42] and found the upregulation of *Cd74* in endocardial cells following left anterior descending artery ligation in the P1 (i.e., regenerative), but not P8 (i.e., non-regenerative), stage (Supplementary Table 2). This observation further suggests that endocardial *Cd74* expression responds to cardiac injury in regenerative settings across species.

We then analyzed an extensive scRNA-seq dataset comprising all cells from regenerating zebrafish hearts[43] and observed a steady increase in the expression of antigen presentation genes within the whole EdC population following cryoinjury until at least 30 dpci (Supplementary Fig. 8a). While classical APCs (i.e., macrophages and B cells) were the main cell groups expressing these antigen presentation genes, we found that other cells, including endothelial cells, also expressed them, although at a lower level (Supplementary Fig. 8b). From this analysis, EdCs as a whole were not amongst the cell groups highly expressing these antigen presentation genes (Supplementary Fig. 8b). Therefore, we asked whether specific EdC populations upregulated their expression of antigen presentation genes, in line with the above-described tissue and EdC-specific gene expression observations (Fig. 1). To address this question, we subclustered EdCs from the referred dataset[43] (Fig. 2a). Out of 12 identified EdC subclusters (Fig. 2b), at least subcluster 10 displayed enriched *cd74a*, *cd74b*, and *mhc2a* expression (Fig. 2c). Interestingly, most of the top marker genes of this subcluster are immune-related (Fig. 2d; Supplementary Data 2), suggesting that these cells acquire an immune-like transcriptomic signature beyond merely activating MHC class II antigen presentation genes. Consistent with the above-described histological and expression data analyses (Fig. 1), the proportion of cells in subcluster 10 increased in regenerating hearts compared with uninjured samples (Fig. 2e). Moreover, the average expression of antigen presentation genes within this subcluster (Fig. 2f) followed a similar pattern as that observed by RT-qPCR analysis of sorted endothelial cells and whole ventricles (i.e., an initial decrease shortly after cryoinjury followed by an increase later on; Fig. 1f–h; Supplementary Fig. 7a–c). Altogether, these findings suggest a potential role for EdCs in the immune response during cardiac regeneration.

### Cd4⁺ T cells accumulate within the injured tissue in close physical proximity to endocardial cells

In our scRNA-seq dataset of sorted immune cells, we identified a small *il4*-expressing cell cluster, likely composed of T cells (Supplementary Figs. 2b and 3b), based on a comparison with the scRNA-seq dataset of whole regenerating ventricles[43]. This cluster could not be fully characterized due to limitations from the small sample size and poor detection of low expressed genes. However, previously published single-cell transcriptomic[43] and histological[27] data have shown the presence of T cells in the regenerating zebrafish heart, altogether

indicating that the antigen presentation-adaptive immunity axis indeed becomes activated during zebrafish cardiac regeneration. Given the increased MHC class II antigen presentation gene expression during zebrafish cardiac regeneration, we next sought to specifically characterize the recruitment of Cd4⁺ T cells (i.e., T helper cells) in the cryoinjured zebrafish heart. Taking advantage of the scRNA-seq dataset of whole regenerating ventricles[43], we first verified that the expression of *cd4-1* was highly specific to T cells and not to other cell types, including other immune cells (Supplementary Fig. 9a). We then used the *TgBAC(cd4-1:mCherry)* line[44] and observed that, while mostly absent in uninjured hearts (Supplementary Fig. 9b) and scarce at 24 hpci, *cd4-1*:mCherry⁺ cells started to accumulate within the injured tissue by 72 hpci and peaked at around 7 dpci (Fig. 3a, b; Supplementary Table 3). Interestingly, we found that these Cd4⁺ T cells were mostly located in close proximity to activated EdCs, as assessed by immunostaining for Aldh1a2 (Fig. 3c). At the analyzed time points, this physical proximity seemed to occur for most, if not all, Cd4⁺ T cells and took place as early as 72 hpci, when only a few Cd4⁺ T cells are present in the injured tissue (Fig. 3c). These observations suggest a crosstalk between activated EdCs and Cd4⁺ T cells.

In summary, during zebrafish cardiac regeneration, antigen presentation is carried out by immune cells and also possibly by activated EdCs. This activation of antigen presentation genes and the recruitment of Cd4⁺ T cells during the regeneration process further point to a role for the adaptive immune system.

### *cd74a; cd74b* mutants display an impaired immune response to cardiac cryoinjury

We next wanted to test this hypothesis that the antigen presentation-adaptive immunity axis plays a role during zebrafish cardiac regeneration. MHC class II antigen presentation relies on multiple proteins including CD74 which allows the assembly and trafficking of MHC class II molecules to present antigens to CD4⁺ T cells[36] (Fig. 4a). Therefore, we targeted MHC class II antigen presentation to investigate its role during the regenerative process by generating mutants for both zebrafish *cd74* paralogs, *cd74a* and *cd74b*. Both mutations consist of small deletions within the region encoding the MHC2 interaction domain (i.e., Δ5 for *cd74a* and Δ25 for *cd74b*), thus predicted to lead to truncated proteins (Fig. 4b, c). Due to the high sequence similarity between Cd74a and Cd74b (Supplementary Fig. 10), we anticipated functional compensation in single mutants, and thus, all downstream analyses were performed in *cd74a; cd74b* double mutants. The levels of both *cd74a* and *cd74b* mutant mRNA in double mutant ventricles were strongly reduced compared with wild type at 72 hpci and 7 dpci (Fig. 4d, e), time points characterized by high levels of expression of antigen presentation genes and Cd4⁺ T cell infiltration, suggesting an efficient loss-of-function at relevant stages. While we did not observe a change in *mhc2a* mRNA levels (Fig. 4f), the lymphoid gene *recombination activating gene 2* (*rag2*) was significantly downregulated in double mutants at both time points (Fig. 4g), suggesting an impaired adaptive immune response. Interestingly, we found that although the

numbers of *cd4-1*:mCherry⁺ T cells were not significantly different between genotypes (Fig. 4h, i), Cd4⁺ T cell infiltration within the central part of the injured tissue was significantly reduced in *cd74a; cd74b* mutants as compared with wild type (Fig. 4h, j; Supplementary Fig. 11).

Physical proximity between activated EdCs and Cd4⁺ T cells did not appear to be affected (Fig. 4k).

To further validate this model of compromised antigen presentation, we performed bulk RNA-seq of ventricles from wild-type

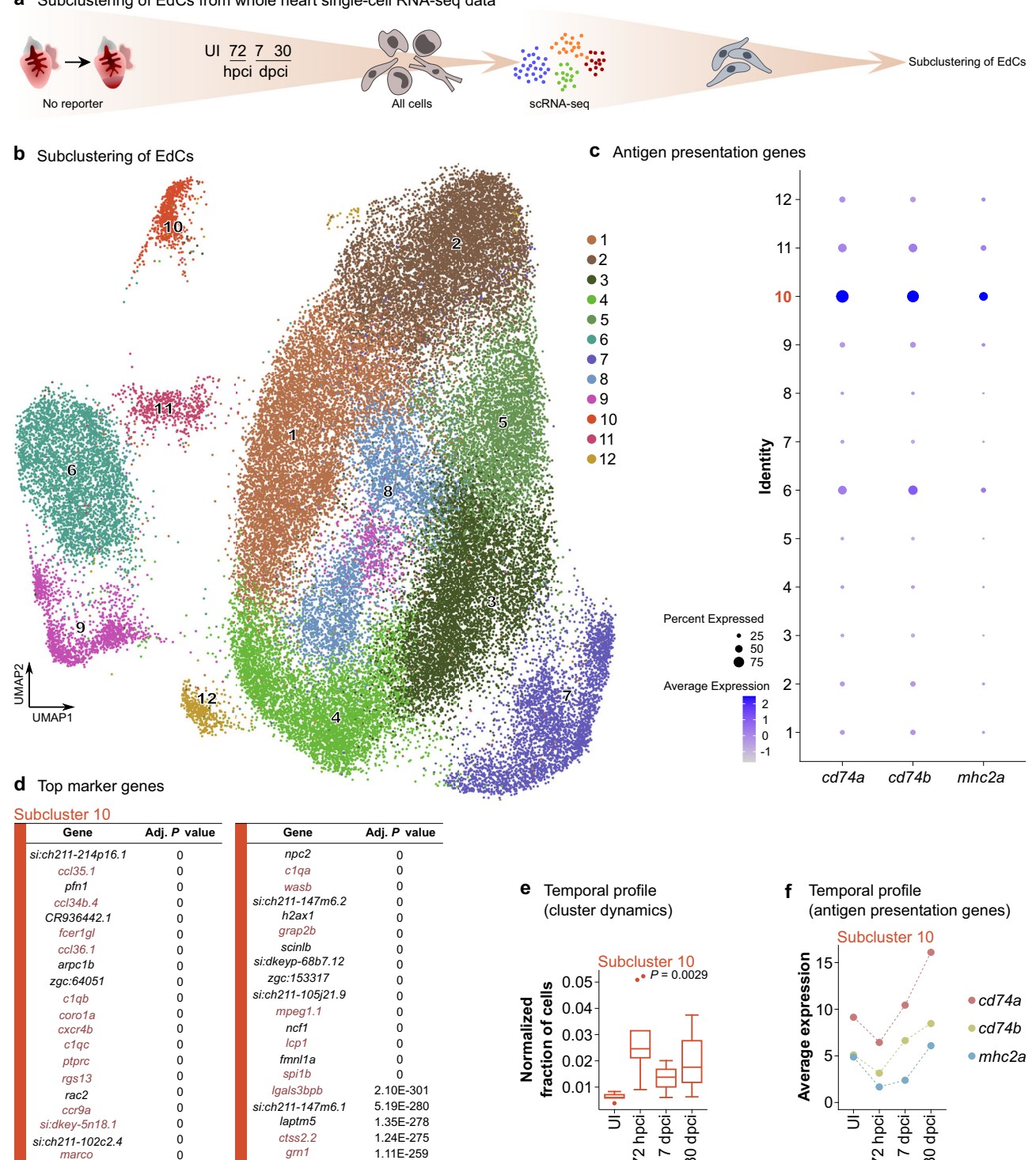

**Fig. 2 | One endocardial cell subcluster exhibits an antigen presentation and immune-like transcriptome. a** Experimental design for subclustering analysis of EdCs from a published scRNA-seq dataset of the whole regenerating zebrafish heart[43]. **b** UMAP plot showing 12 different EdC subclusters. **c** Dot plot showing enriched expression of antigen presentation genes in at least subcluster 10. **d** List of top 40 marker genes in subcluster 10, showing strong enrichment in immune-related genes (red). Wilcoxon Rank Sum Test; adjusted *P* value based on Bonferroni

correction. **e** Box plot showing the increased relative abundance of cells in subcluster 10 during regeneration compared with uninjured; box plot represents the median, Q1, Q3, the minimum, and the maximum; dots represent potential outliers; *n* = 5, 10, 13, 5 independent datasets for UI, 72 hpci, and 7 and 14 dpci, respectively (see[43]); one-way ANOVA (*P* value included in the graph). **f** Plot showing the average expression of antigen presentation genes within subcluster 10 at different time points, revealing increased levels during regeneration. UI uninjured.

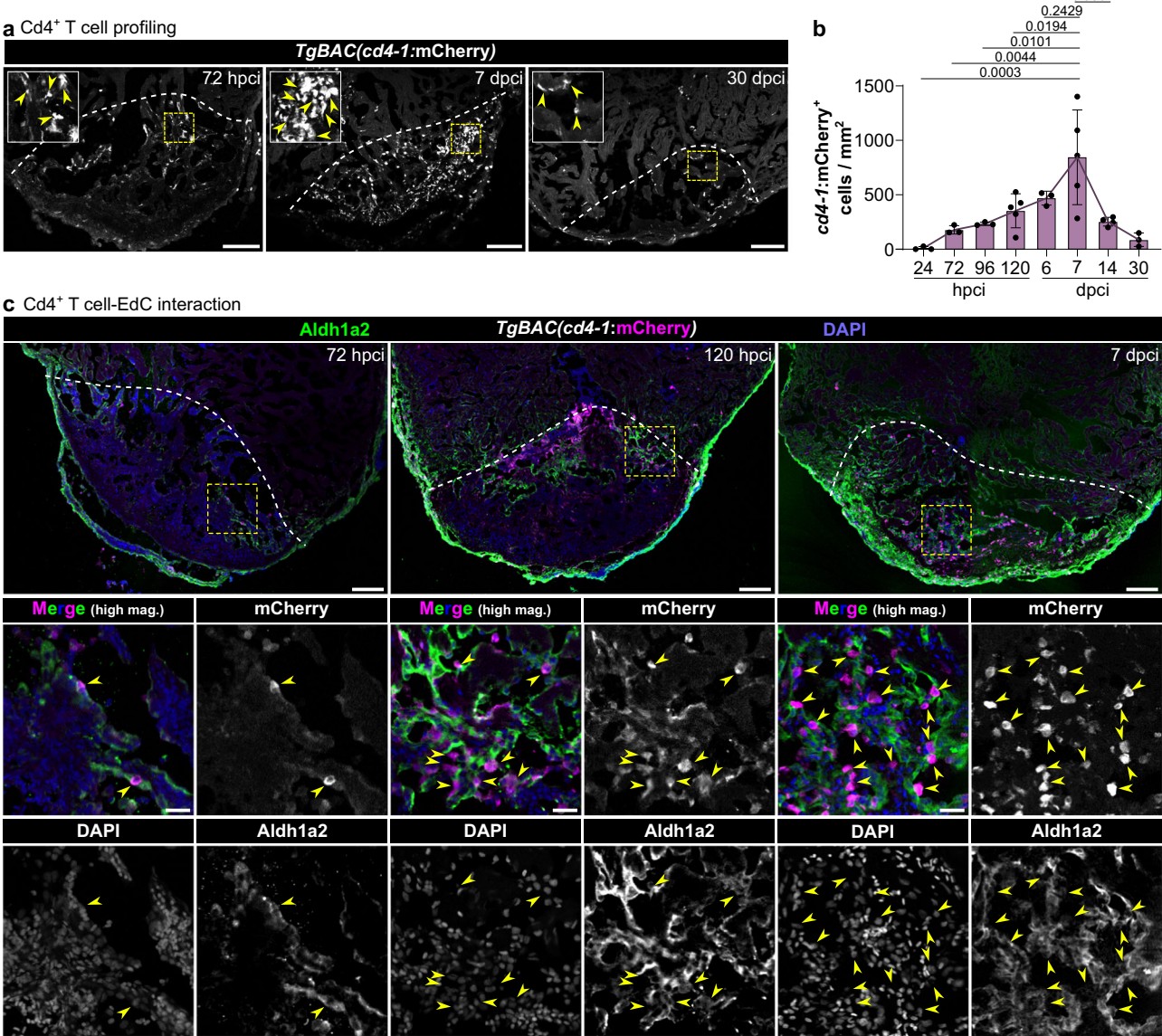

**Fig. 3 | Cd4⁺ T cells are present in the injured cardiac tissue and associate with the activated endocardium. a** Confocal images of representative cryosectioned ventricles from adult *TgBAC(cd4-1:mCherry)* zebrafish at various time points after cryoinjury, immunostained for mCherry, showing Cd4⁺ T cells in the injured tissue (yellow arrowheads). **b** Quantification of total *cd4-1*:mCherry⁺ cells within the injured tissue, showing a peak at 7 dpci. Dots in the graph represent individual ventricles, and the bars represent the mean ± SD; $n = 3$ (24 hpci), 3 (72 hpci), 3 (96 hpci), 5 (120 hpci), 3 (6 dpci), 5 (7 dpci), 4 (14 dpci), and 3 (30 dpci) biologically independent samples; one-way ANOVA ($P = 0.0003$) and Tukey's post hoc test for multiple comparisons; *P* values included in the graph; full list of multiple comparisons in Supplementary Table 3. **c** Confocal images of representative cryosectioned ventricles from adult *TgBAC(cd4-1:mCherry)* zebrafish at various time points after cryoinjury, immunostained for mCherry and Aldh1a2, revealing the close association of Cd4⁺ T cells (yellow arrowheads) with activated EdCs (green). Yellow dashed squares outline the magnified areas; dashed lines mark the border of the injured tissue. Scale bars: 100 μm in (**a**) and (**c**) (low magnification); 20 μm in **c** (high magnification).

and *cd74a; cd74b* mutant ventricles at 120 hpci (i.e., during the activation of antigen presentation genes in EdCs and immune cells and Cd4⁺ T cell infiltration in the injured tissue) (Fig. 5a). Amongst the top downregulated genes in the mutant ventricles (Fig. 5b; Supplementary Fig. 12a), at least 35% have known or predicted immune functions (Supplementary Fig. 12b). Searching this dataset led us to identify several other downregulated genes with well-known immune functions, including the regulation of myeloid cell development (e.g., *csf1rb, csf1ra, csf3r*), the regulation of the inflammatory response (e.g., *nfkb1, rela, grn1, grn2, grna*), and the regulation of antigen presentation (e.g., *cd40*, which encodes a costimulatory molecule for MHC class II antigen presentation) (Fig. 5c). Notably, we observed the dysregulation of genes related to T cell activation and survival (Fig. 5d).

*cd4-1* expression was lower in mutants compared with wild type, consistent with the reduced Cd4⁺ T cell infiltration in the injured tissue. Similarly, expression of *bcl2a*, a gene that functions to suppress lymphoid cell apoptosis[45], was lower in mutants compared with wild type. In contrast, expression of *mafb* (an ortholog of c-MAF, an inhibitor of *BCL2* expression and regulator of T cell function[46,47]) was higher in mutants compared with wild type. While only the PPAR signaling pathway was found to be significantly upregulated in *cd74a; cd74b* mutant ventricles, gene enrichment analyses identified the downregulation of several immune related biological processes and pathways (Fig. 5e–g), as well as the downregulation of the VEGF and ErbB signaling pathways (Fig. 5g), which have been implicated in cardiac regeneration[48–50].

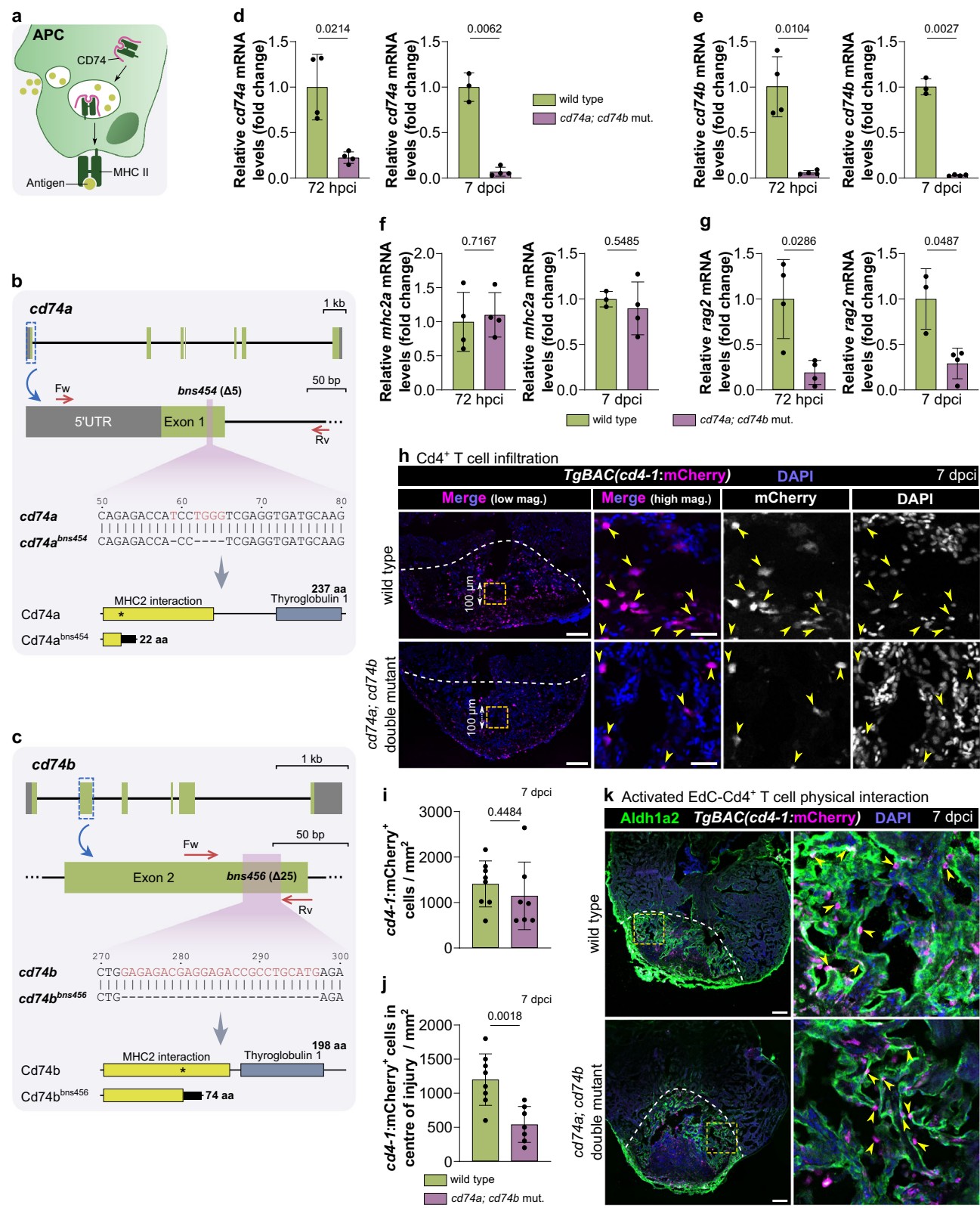

Altogether, these data show that Cd74 is required for the adaptive immune response in the regenerating zebrafish heart.

### cd74a; cd74b mutants exhibit defects in the activated endocardium

We further observed that the MAPK signaling pathway was also downregulated in the double mutant ventricles (Fig. 5g). MAPK signaling has previously been reported to be essential for zebrafish cardiac regeneration and to be associated with the activated endocardium in the ventricular resection model[51]. Here, we used phospho-ERK (pERK) as a readout for MAPK signaling and confirmed its presence in Aldh1a2[+] activated EdCs at 7 dpci (Fig. 6a; Supplementary Fig. 13a). We also confirmed that Aldh1a2 accumulates in cells displaying nuclear Fli1 immunostaining, another indicator of EdCs

**Fig. 4 | *cd74a; cd74b* mutants exhibit reduced Cd4⁺ T cell infiltration of the injured cardiac tissue. a** Schematic showing the involvement of CD74 in MHC class II antigen presentation. **b, c** Schematics of the *cd74a* (**b**) and *cd74b* (**c**) wild-type and mutant alleles (bns454 and bns456) depicting the wild-type gene structure, location of the mutations, primers used for genotyping, nucleotide deletions (red) and the predicted resulting proteins, highlighting the different domains. The numbers mark the nucleotide positions in the respective coding sequences. Asterisks mark the mutation site. **d–g** Relative mRNA levels of *cd74a* (**d**), *cd74b* (**e**), the antigen presentation gene *mhc2a* (**f**) and the lymphoid gene *rag2* (**g**) in wild-type and *cd74a; cd74b* mutant ventricles at 72 hpci and 7 dpci. Ct values included in Supplementary Table 7. **h** Confocal images of representative cryosectioned ventricles from adult *TgBAC(cd4-1:mCherry)* wild-type and *cd74a; cd74b* mutant zebrafish at 7 dpci, immunostained for mCherry, showing a reduction in the infiltration of Cd4⁺ T cells (yellow arrowheads) in the central part of the injured tissue in double mutants; yellow dashed squares outline the magnified areas shown on the right and these areas were used to quantify the data depicted in **j**; dashed lines mark the border of the injured tissue. **i, j** Quantification of total *cd4-1*:mCherry⁺ cells within the whole (**i**) or central (**j**) region of the injured tissue of wild types and *cd74a; cd74b* mutants. Dots in the graphs represent individual ventricles, and the bars represent the mean ± SD; *n* = 4 (wild type and *cd74a; cd74b* mut.) biologically independent samples for 72 hpci and 3 (wild type) and 4 (*cd74a; cd74b* mut.) for 7 dpci in **d–g**, and 8 (wild type) and 7 (*cd74a; cd74b* mut.) biologically independent samples in **i, j**; two-tailed Welch's *t* test (*P* values included in the graphs). **k** Confocal images of representative cryosectioned ventricles from adult *TgBAC(cd4-1:mCherry)* wild-type and *cd74a; cd74b* mutant zebrafish at 7 dpci, immunostained for mCherry and Aldh1a2, showing apparently unchanged physical proximity between activated EdCs and Cd4⁺ T cells. Scale bars: 100 μm (low magnification); 20 μm (high magnification). Green and gray boxes in **b** and **c** represent exons and UTRs, respectively. aa amino acids, MHC II Major Histocompatibility Complex Class II, Fw forward primer, Rv reverse primer, kb kilobase pairs, bp base pairs.

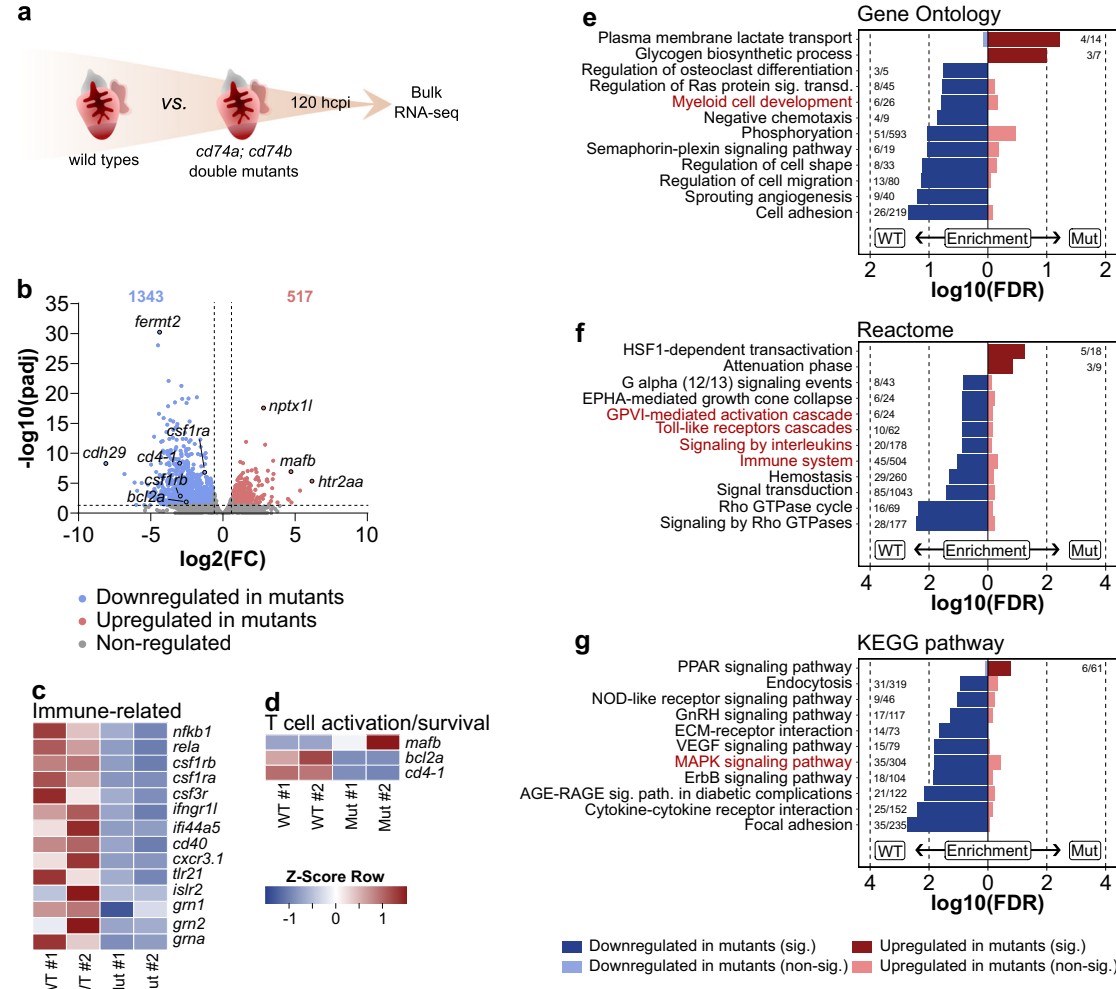

**Fig. 5 | The immune response is hampered in *cd74a; cd74b* mutants.**
**a** Experimental design for bulk RNA-seq analysis of injured ventricles from wild types and *cd74a; cd74b* mutants at 120 hpci. **b** Volcano plot showing differentially expressed genes (FC ≥ 2; FDR < 0.05). Blue and red dots represent genes down- and upregulated, respectively, in *cd74a; cd74b* mutants compared with wild types. Blue and red values represent the number of significantly down- and upregulated genes, respectively; Wald test, corrected for multiple testing using the Benjamini-Hochberg method. **c, d** Heat maps showing the relative expression of genes encoding known immune-related factors (**c**), and proteins involved in T cell activation and survival (**d**) in individual samples. **e–g** Gene enrichment analyses showing significantly regulated Gene Ontology terms (**e**), Reactome (**f**) and KEGG pathways (**g**); immune-related processes and pathways marked in red. FDR false discovery rate, FC fold change, WT wild-type samples, Mut *cd74a; cd74b* mutant samples.

(Supplementary Fig. 13b), consistent with previous reports[52–54]. Importantly, as compared with wild-type, *cd74a; cd74b* mutant ventricles displayed defects in the regenerating endocardium at 7 dpci, characterized by a reduction in the area covered by the endocardial-specific pERK signal (Fig. 6b, c). In addition, we found that in *cd74a;*

*cd74b* mutants, the pERK⁺ activated endocardium failed to efficiently populate deeper regions of the injured tissue (Fig. 6b, d), coinciding with the findings on the reduced Cd4⁺ T cell infiltration within the injured tissue (Fig. 4h, j). In addition, EdU labeling experiments, together with immunostaining for the endocardial marker Wif1[55],

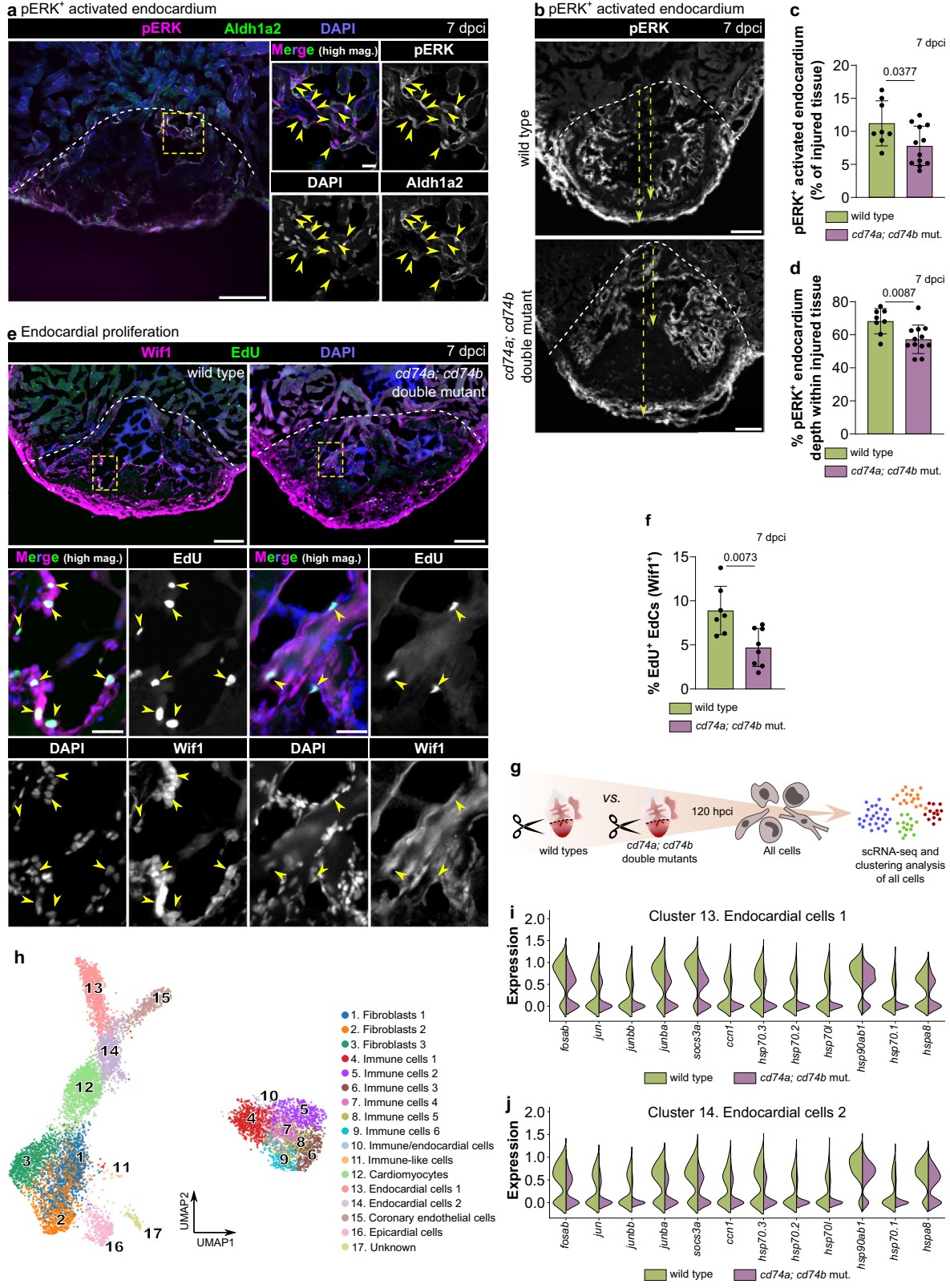

revealed a marked reduction in EdC proliferation within the injured tissue (Fig. 6e, f; Supplementary Fig. 13c).

To better understand the cell type-specific alterations in *cd74a; cd74b* mutants, we carried out scRNA-seq of wild-type versus double mutant ventricles at 120 hpci (i.e., during upregulated antigen presentation gene expression and at the peak of Cd4+ T cell infiltration, as well as shortly before the observed endocardial phenotype) (Fig. 6g;

Supplementary Table 4). To perform an unbiased analysis and enrich for injury-specific cells, we used all the cells from the injured and border zone tissues. After quality control and filtering, the final dataset comprised 10827 cells (3598 and 7229 from wild-type and mutant samples, respectively) grouped in 17 major clusters. Major cardiac cell types were identified, including cardiomyocytes, fibroblasts, epicardial cells, coronary endothelial cells, various immune cell subsets, and 2

**Fig. 6 | Endocardial regeneration is compromised in *cd74a; cd74b* mutants.**
**a** Confocal images of a representative cryosectioned adult zebrafish ventricle at 7 dpci, immunostained for pERK and Aldh1a2, showing pERK accumulation in EdCs (yellow arrowheads). Two independent experiments with similar results.
**b–d** Confocal images of representative cryosectioned adult zebrafish ventricles at 7 dpci (**c**), and quantification of the percentage of pERK⁺ area within the injured tissue (**c**) and of the depth of pERK⁺ cells into the injured tissue (**d**), showing an overall reduction in *cd74a; cd74b* mutants compared with wild types. Yellow dashed arrows in (**b**) show the two distances measured to calculate the occupancy of pERK⁺ EdCs within the injured tissue quantified in (**d**). **e, f** Images of representative cryosectioned ventricles from 7 dpci EdU-treated adult zebrafish immunostained for Wif1 (**e**) and quantification of proliferating Wif1⁺ EdCs in the injured tissue (**f**; arrowheads in **e**). **g** Experimental design for scRNA-seq analysis of all cells from the border zone

and injured tissue of wild types and *cd74a; cd74b* mutants at 120 hpci. **h** UMAP plot showing the clustering analysis of the scRNA-seq data containing all wild-type and *cd74a; cd74b* mutant cells. **i, j** Violin plots of representative genes revealing an overall reduction in the stress response of *cd74a; cd74b* mutant EdCs in both clusters 13 (**i**) and 14 (**j**). Full list of differentially expressed genes and statistical information can be found in Supplementary Data 3. Dots in the graphs represent individual ventricles, and the bars represent the mean ± SD; *n* = 8 (wild type) and 12 (*cd74a; cd74b* mut.) biologically independent samples in **c, d** and 7 (wild type) and 8 (*cd74a; cd74b* mut.) biologically independent samples in (**f**); two-tailed Welch's *t* test (*P* values included in the graphs). Yellow dashed square and rectangles outline the magnified areas; dashed lines mark the border of the injured tissue. Scale bars: 100 μm (low magnification); 20 μm (high magnification).

endocardial cell clusters (Fig. 6h; Supplementary Fig. 14a–c). We also identified a small cluster (i.e., cluster 10) that expresses both immune (e.g., antigen presentation) and endothelial/endocardial (e.g., *kdrl*, *fli1a*, *mb*, *spock3*, *aqp8a.1*, *aldh1a2*) genes (Supplementary Fig. 14b). This cluster likely represents, at least partially, the antigen presentation gene-expressing EdCs described above. Given the relatively low transcriptome coverage per cell in this dataset, we cannot exclude the possibility that cells within the other two identified endocardial clusters also express these antigen presentation genes. Importantly, this scRNA-seq dataset further allowed us to identify, in both endocardial cell clusters (i.e., clusters 13 and 14), the reduced expression of several genes involved in the stress response, including AP-1 factor genes (e.g., *fosab*, *jun*, *junbb*, *junba*) (Fig. 6i, j; Supplementary Data 3), further suggesting impaired endocardial function.

Altogether, these data indicate that Cd74 function is required for the increased presence of EdCs in the injured tissue, possibly by promoting EdC proliferation.

## Cardiomyocyte repopulation is hampered in *cd74a; cd74b* mutants after cardiac cryoinjury

The formation of new myocardium, which in zebrafish involves cardiomyocyte dedifferentiation and proliferation[23,56,57], is part of an efficient regenerative response. To determine whether *cd74a; cd74b* mutants display additional regenerative defects, we first immunostained cryoinjured hearts with the N2.261 antibody to mark dedifferentiated (and undifferentiated) cardiomyocytes[58]. We found that at 120 hpci and compared with wild type, *cd74a; cd74b* mutants exhibit a pronounced (i.e., almost 50%) and statistically significant reduction in the percentage of N2.261⁺ cardiomyocytes in the border zone, indicative of decreased cardiomyocyte dedifferentiation (Fig. 7a, b). We also observed at 7 dpci a 50% reduction, also statistically significant, in cardiomyocyte proliferation in the border zone, as assessed by EdU incorporation (Fig. 7c, d). We next explored our 120 hpci border zone and injured tissue scRNA-seq dataset (Fig. 6h,i) and observed decreased expression of genes related to cardiac function and cardiomyocyte dedifferentiation (Fig. 7e; Supplementary Data 4). Notably, we found that several glycolytic enzymes also displayed reduced expression in *cd74a; cd74b* mutants (Fig. 7e; Supplementary Data 4), pointing to defective glycolysis, a key metabolic feature of cardiomyocyte regeneration[59]. Further analysis allowed the identification of 9 cardiomyocyte subclusters (Supplementary Fig. 15a). The cardiomyocytes in subcluster 9 display a more immature state, with a higher expression of genes such as *nppb*, *tnnt2a* and *mustn1b*[23,60] (Supplementary Data 5), and exhibit a tendency to be less numerous in *cd74a; cd74b* mutants than in wild types (Supplementary Fig. 15b; Supplementary Table 5). Altogether, these data indicate that Cd74 function and antigen presentation are required for efficient cardiomyocyte repopulation. In a parallel approach, we exposed wild-type zebrafish to cyclosporine A (CsA), which has been shown to block T cell activation and function in mammals[61,62] and has been previously used in adult zebrafish to inhibit Calcineurin[63]. We performed daily intraperitoneal

injections from 72 hpci to 6 dpci (Supplementary Fig. 16a) and observed that this treatment resulted in a tendency towards a decreased number of injury-infiltrating Cd4⁺ T cells at 7 dpci (Supplementary Fig. 16b, c). Using a dedifferentiated cardiomyocyte reporter line *(Tg(gata4:EGFP))*[56], we observed a strong reduction in transgene expression in cryoinjured hearts from CsA-treated zebrafish compared with controls (Supplementary Fig. 16d). In addition, cardiomyocyte proliferation was reduced upon CsA treatment to a similar extent as in *cd74a; cd74b* mutants (Supplementary Fig. 16e, f). Thus, CsA treatment led to similar cardiomyocyte defects as observed in *cd74a; cd74b* mutants. While we cannot exclude additional effects of Calcineurin inhibition, these data support the model that Cd74 promotes cardiomyocyte repopulation through the modulation of T cell activation and function.

Progressive resolution of transient collagen accumulation in the injured tissue is another hallmark of cardiac regeneration in zebrafish[16,24,64]. However, we found that the scar areas were not statistically different between wild types and *cd74a; cd74b* mutants at 60 dpci (Fig. 7f, g; Supplementary Figs. 17 and 18). At this stage, myocardial wall formation and fibrin resolution also did not appear to differ between genotypes (Supplementary Fig. 19). We did observe tissue constrictions close to the injury site more frequently in *cd74a; cd74b* mutants compared with wild type (Fig. 7f, h; Supplementary Figs. 17 and 18), although the reason for, and significance of, these constrictions are unclear at this time.

Altogether, these data show the requirement of Cd74 for cardiomyocyte repopulation, but not for the resolution of collagen accumulation in the injured tissue.

## Discussion

The immune system, including the adaptive immune system, may be key to promote regenerative responses post-injury[14,65]. Studying regenerative models, and identifying the differences with non-regenerative scenarios, has been instrumental to uncover the roles of the immune system in driving heart regeneration[28]. Yet, much remains to be understood in terms of how different immune cells and immune processes modulate regenerative responses. Here, using the zebrafish, a regenerative model, we investigated the participation of the antigen presentation-adaptive immunity axis in adult cardiac regeneration.

While all nucleated cell types are able to present antigens via MHC class I[66,67], MHC class II antigen presentation is typically associated with immune cells such as macrophages, dendritic cells, and B cells[11,67,68]. In our study, we found that activated EdCs, a cardiac-restricted form of endothelial cells, also activate MHC class II antigen presentation genes after cardiac injury. In addition, activated EdCs are in close physical proximity to Cd4⁺ T cells in the injured tissue, pointing to a crosstalk between these two cell types. Endothelial cells have been reported to express basal levels of MHC class II genes[69–71]. Lymphatic endothelial cells in particular, have also been shown to express MHC class II genes[72–74]. In addition, tumor lymphatic endothelial cells have been

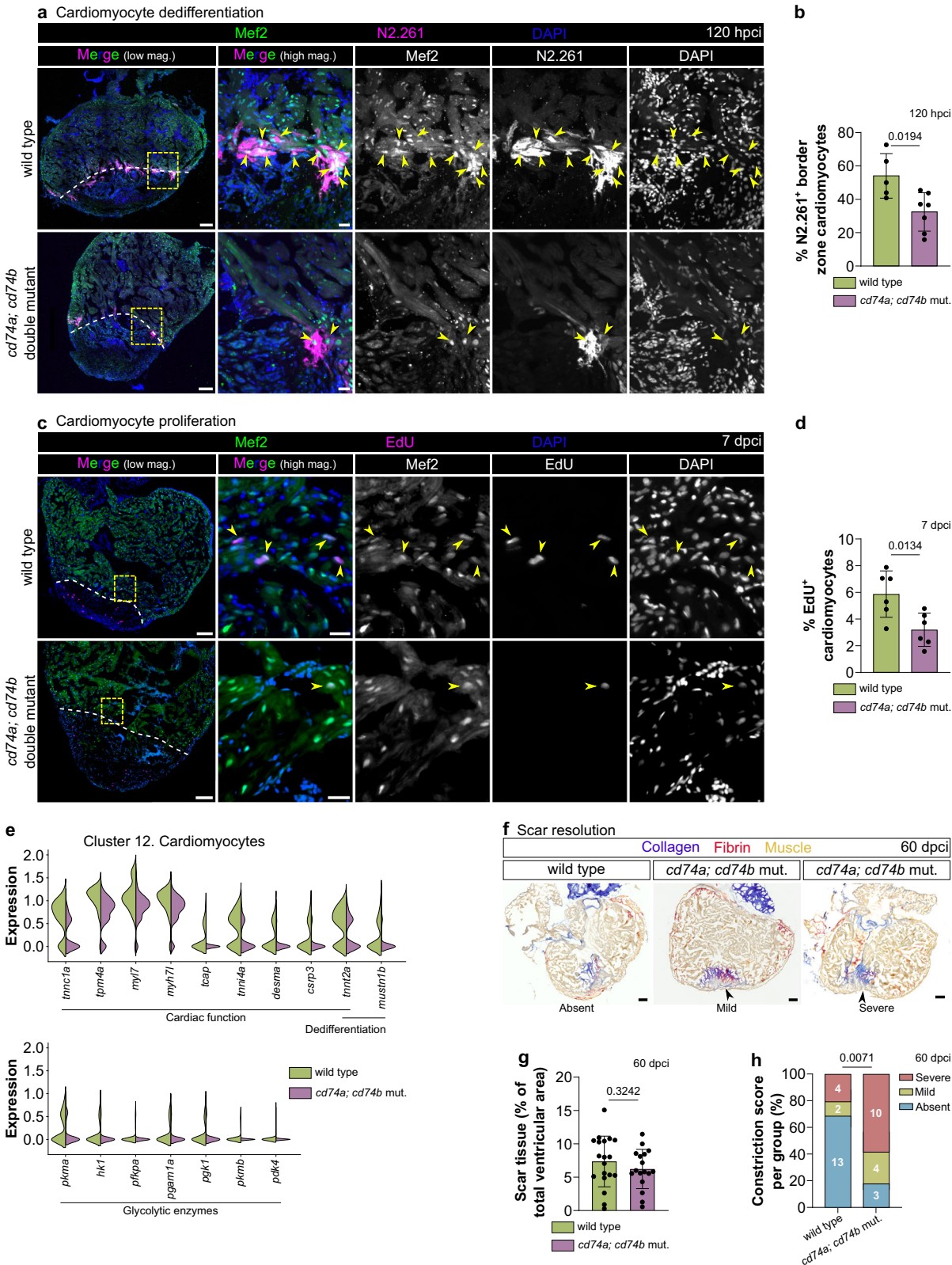

shown to function as APCs in antitumor immunity[74]. However, the expression of MHC class II genes in EdCs is mostly uncharacterized, with one study reporting such expression in dilated cardiomyopathy patients[75]. A recent report has identified immune responsive and regulatory gene signatures in cardiac endothelial cells during mouse heart regeneration and in human heart failure samples[76]. Accordingly, we also found that in addition to the upregulation of antigen presentation genes, these activated EdCs appear to exhibit an immune-like transcriptome. Whether this is a transient feature or some EdCs acquire a more immune-like phenotype, is unclear. Indeed, the endocardium is known for its high level of plasticity, contributing to numerous cell types in various conditions, including during regeneration[77]. For example, in the developing mouse heart, some EdCs function as a source of hematopoietic progenitors, constituting the hemogenic

**Fig. 7 | Cardiomyocyte regeneration is compromised in *cd74a; cd74b* mutants.**
**a**, **b** Images of representative cryosectioned adult zebrafish ventricles at 120 hpci, immunostained for Mef2 and N2.261 (**a**), showing dedifferentiating cardiomyocytes (yellow arrowheads), and respective quantification (**b**), showing a significant decrease in *cd74a; cd74b* mutants. **c**, **d** Images of representative cryosectioned ventricles from 7 dpci EdU-treated adult zebrafish, immunostained for Mef2 (**c**), showing proliferating cardiomyocytes (yellow arrowheads), and respective quantification (**d**), showing a decrease in *cd74a; cd74b* mutants. Yellow dashed rectangles outline the magnified areas; dashed lines mark the border of the injured tissue. **e** Violin plots of representative genes revealing an overall reduction in the expression of markers of cardiomyocyte function and dedifferentiation and of glycolytic enzymes in *cd74a; cd74b* mutant cardiomyocytes (cluster 12 from scRNA-seq in Fig. 6). Full list of differentially expressed genes and statistical information can be found in Supplementary Data 4. **f** Brightfield images of representative

cryosectioned ventricles at 60 dpci, stained with AFOG, revealing the scar tissue by the collagen staining (blue) and tissue constrictions close to the injured tissue (arrowhead). **g** Quantification of scar size, showing no significant difference between wild type and *cd74a; cd74b* mutants. **h** Quantification of tissue constriction index, showing increased severity in *cd74a; cd74b* mutants compared with wild type. Scoring categories are exemplified in **f**. *n* = 19 wild types and 17 *cd74a; cd74b* mutants; the graph in **h** shows the percentage calculated based on the observed frequency; white numbers represent the counts per category. Dots in the bar graphs represent individual ventricles, and the bars represent the mean ± SD; *n* = 5 (wild type) and 7 (*cd74a; cd74b* mut.) biologically independent samples in (**b**) and 6 (wild type and *cd74a; cd74b* mut.) biologically independent samples in (**d**); two-tailed Welch's *t* test (*P* values included in the graphs) in **b**, **d**, **g**; Fisher's exact test in (**h**), using observed frequency values. Scale bars: 100 μm (low magnification); 20 μm (high magnification).

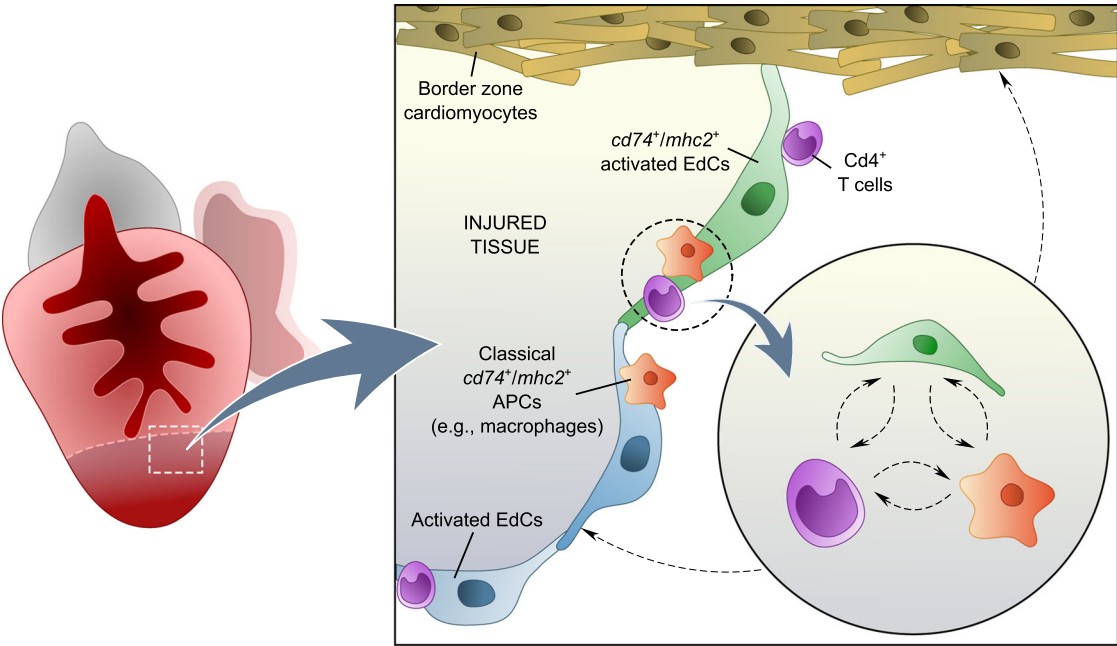

**Fig. 8 | Proposed model for the role of MHC class II antigen presentation during zebrafish cardiac regeneration.** The model depicts the crosstalk between antigen presentation gene-expressing EdCs, Cd4⁺ T cells, and classical APCs, including macrophages. The activity of these cells promotes endocardial and myocardial regeneration.

endocardium[78]. Another study from the same group subsequently reported that the endocardium also contributes to macrophages in the developing cardiac valves[79]. Interestingly, these endocardial-derived macrophages play a role in antigen presentation. In line with these reports, we also found that neonatal mouse EdCs upregulate *Cd74* expression in a regenerative stage, but not in a non-regenerative one, further suggesting its role in this process during cardiac regeneration.

In our study, we aimed to investigate the function of antigen presentation during adult cardiac regeneration. While the direct targeting of MHC class II genes is likely an effective strategy, the high number of these genes in zebrafish constitutes an additional challenge. Therefore, we decided to target Cd74, which, as introduced earlier, is required for the assembly and trafficking of MHC class II molecules, and found that loss of Cd74 function indeed results in a compromised immune response. Specifically, *cd74a; cd74b* mutants display impaired infiltration of Cd4⁺ T cells into the injured cardiac tissue. In addition, they exhibit decreased EdC occupancy in the injured tissue, as well as a reduction in EdC proliferation. These EdC defects are potentially direct (i.e., antigen presentation gene expression in EdCs has a direct effect on their behavior) or indirect (i.e., impaired adaptive immunity caused by compromised antigen presentation affects EdC behavior). In

addition, cardiomyocyte dedifferentiation at 120 hpci and proliferation at 7 dpci are also reduced in *cd74* mutants. We hypothesize that EdCs, Cd4⁺ T cells, and/or classical APCs signal to cardiomyocytes. During cardiogenesis, the endocardium is known to signal to cardiomyocytes[80,81]. In the regenerating zebrafish heart, EdCs have also been reported to signal to cardiomyocytes through pathways such as Notch to regulate their dedifferentiation and proliferation[20,52,55,82]. In addition, T_regs, which can also be Cd4⁺, secrete the cardiomyocyte mitogen Neuregulin 1 during zebrafish heart regeneration[37], thereby directly promoting cardiomyocyte proliferation. T_regs have also been reported to stimulate cardiomyocyte proliferation in the injured adult[83] and neonatal[84] mouse heart. Moreover, macrophages are key APCs that interact with Cd4⁺ T cells[11], and they have been reported to promote cardiomyocyte proliferation during larval[85] and adult[30] zebrafish heart regeneration. Although CD74 is required for MHC class II antigen presentation, it can also function, at least in mammals, as a macrophage receptor for Macrophage migration inhibitory factor in certain conditions[36], including post-tissue injury[86], thereby also linking the innate and adaptive immune systems. Altogether, our data support a model in which a potential crosstalk between activated EdCs, Cd4⁺ T cells, and/or macrophages is required to promote cardiomyocyte repopulation, as well as endocardial regeneration (Fig. 8).

The roles of the adaptive immune system, in particular the mechanisms activated by MHC class II antigen presentation, following cardiac injury remain a matter of debate. While CD4[+] T cells have been reported to be required for wound healing and survival post-MI in mice[10], other reports point to their detrimental role in the infarcted heart[8,9]. In zebrafish, a study has shown that global T cell populations are recruited to the regenerating heart[27], pointing to a cardioprotective role. Here, we show the recruitment of Cd4[+] T cells and reveal the requirement of MHC class II antigen presentation for several aspects of zebrafish adult heart regeneration, further illustrating a pro-regenerative role for the adaptive immune response. The complex diversity of T cell populations may explain the differences in post-cardiac injury outcome. In fact, repeated cryoinjuries of the zebrafish heart result in decreased regenerative performance[87], a phenotype we speculate could be associated with the increasing enrichment of autoreactive T cell clones. Another study has found that the adoptive transfer of adult T cells impaired cardiac regeneration in neonatal mice[88]. Together, these findings suggest the hypothesis that the balanced activity between adverse and regenerative adaptive immune programs, which differs between models, modulates the regenerative capacity of the heart. Identifying such programs is essential to define approaches to block adverse T cell responses and/or stimulate regenerative ones. Self-antigens play a major role in driving adaptive immune responses post-MI, but only a few have been identified thus far. Myosin heavy chain α has been shown to be a predominant self-antigen in the activation of the CD4[+] T-cells that acquire a $T_{reg}$ phenotype with cardioprotective properties post-MI[89]. Recently, a peptide fragment of the beta-1 adrenergic receptor (ADRB1) has been found to stimulate CD4[+] T cell responses[90]. Damage-associated molecular patterns (DAMPs) have also been described to activate APCs (e.g., dendritic cells) which can then polarize naïve T cells[91]. Ultimately, understanding the antigen presentation mechanisms and identifying the various antigens at play following cardiac injury in regenerative and non-regenerative settings should provide an important basis for immunomodulation strategies.

## Methods

### Zebrafish handling and husbandry
Animal husbandry followed standard conditions in accordance with institutional (Max-Planck-Gesellschaft) and national ethical and animal welfare guidelines approved by the ethics committee for animal experiments at the Regierungspräsidium Darmstadt, Germany. All efforts were made to minimize pain, distress, and discomfort. Anesthesia was performed by incubating zebrafish in 0.6 mM tricaine solution (MS-222; Sigma-Aldrich). Euthanasia was performed using a lethal dose of anesthetic. Hearts were collected following euthanasia. For comparison purposes, two sibling batches, one wild-type and one *cd74a; cd74b* double mutant, were crossed with *cd74a; cd74b* double heterozygous animals from the same batch. The generated wild-type and *cd74a; cd74b* double mutant progeny were used. All experiments included females and males: i) for experiments using an even number of animals, 50% of females and 50% of males were chosen for each group; ii) for experiments using an uneven number of animals, 50% of females and 50% of males plus 1 female or 1 male were used. All animals were chosen randomly.

### Zebrafish lines
Wild-type, transgenic and mutant zebrafish (3 to 12 months of age) used in this study were from the AB strain. The following transgenic and mutant lines were used: *Tg(mhc2dab:EGFP)[sd6]* [39], *Tg(ptprc:DsRed)[sd3]* [40], *Tg(mpeg1:EGFP)[gl22]* [92], *Tg(UAS:NTR-mCherry)[c264]* [93], *TgBAC(cd4-1:mCherry)[umc13]* [44], *Tg(gata4:EGFP)[ae1]* [94], *Et(krt4:EGFP)[sqet331AEt]* [41], *Tg(cd74a:Gal4ff; cryaa:mCerulean)[bnsSS2]* (this study), *cd74a[bns454]* (this study), and *cd74b[bns456]* (this study).

### Generation of mutant and transgenic lines
The *cd74a[bns454]* and *cd74b[bns456]* mutants were generated using the CRISPR-Cas9 technology with the following guide RNA (gRNA) sequences (5'–3'): GCAGAGACCATCCTGGGTCG**AGG** (exon 1 of *cd74a*) and GCGGTCTCCTCGTCTCTCCA**GGG** (exon 2 of *cd74b*; PAM sequences in bold). gRNAs were generated by cloning the gRNA sequences into the pT7-gRNA plasmid[95], followed by in vitro transcription using the MegaShortScript T7 Transcription Kit (Thermo Fisher Scientific) and purification using an RNA-cleanup column (Biozym). 75 picograms (pg) of each gRNA was injected into one-cell stage wild-type embryos, together with 150 pg of Cas9 mRNA. Mutations were identified by direct sequencing of the PCR products in F1 zebrafish, following outcrosses of potential founders with wild types. Genotyping of the *cd74a[bns454]* allele was performed through PCR amplification, using the primers fw 5'- TTCCGTCACAGACATCCTGA-3' and rev 5'- CACTGAGCACCAACAAGTTCA-3', followed by digestion with the restriction enzyme XhoI (i.e., with the mutant allele containing the restriction site). Genotyping of the *cd74b[bns456]* allele was carried out through high-resolution melting analysis (HRMA; Eco-Illumina), using the primers fw 5'- TGCTTACATGGCCTACAGTCA-3' and rev 5'- GACCAGATTTACGGCTGAT-3'.

To generate the *Tg(cd74a:Gal4ff; cryaa:mCerulean)* line, hereafter referred to as *Tg(cd74a:Gal4ff)*, a 2.6 kb regulatory region upstream of the *cd74a* transcription start site was cloned together with *Gal4ff* into a Tol2-based destination vector containing a *cryaa* promoter driving mCerulean. Cloning was performed using the In-Fusion HD Cloning Kit (Takara). The *cd74a* promoter was amplified using the primers fw 5'-GGCCGACCCAATCAGAGG-3' and rev 5'-CACTGCTGTGTGTGAGTGTG-3', and the final insertions into the destination vector were confirmed by sequencing. The *cd74a:Gal4ff* construct was injected into one-cell stage embryos together with 50 pg of Tol2 transposase mRNA. These animals were raised to adulthood, outcrossed with wild type, and the progeny was screened for fluorescent eyes and a single F1 animal was used to generate a stable line.

### Cryoinjury
Cardiac damage was induced through ventricle cryoinjury[17]. Briefly, anesthetized animals were placed on a wet sponge with the ventral side up. A small incision was performed in the thoracic region and the ventricle was exposed. Injury was performed by applying a liquid nitrogen cooled probe until thawing. Fish were left to recover from anesthesia in fresh system water.

### Histological analyses and imaging
For histological analyses, ventricles were fixed in 4% PFA for 1 hour at room temperature, washed 3 times in 1× PBS, and preserved overnight with 30% (w/v) sucrose solution prepared in 1× PBS at 4 °C. The samples were then embedded in Tissue-Tek O.C.T. Compound (Sakura Finetek) and kept at −80 °C. Cryosections (10 μm thick) were obtained using a Leica CM3050S or CM1950 cryostat (Leica Microsystems GmbH), collected onto SuperFrost Plus slides (Thermo Fisher Scientific) and stored at −20 °C until further use.

For immunofluorescence, slides were thawed for 15 min at room temperature and washed twice with 1× PBS to remove the O.C.T. Compound. Slides were washed with 0.1 M glycine (Sigma-Aldrich) followed by permeabilization for 7 min at −20 °C in pre-cooled acetone (Carl Roth). To immunostain for Mef2, this last step was substituted by antigen retrieval in 10 mM sodium citrate buffer pH 6.0 with 0.05% (v/v) Tween (all from Sigma-Aldrich) for 20 min at 95 °C followed by fixation with a 10% hydrogen peroxide (Sigma-Aldrich) and 90% methanol (Carl Roth) solution for 10 min. Sections were then incubated in PBDX blocking solution [1% (w/v) bovine serum albumin, 1% (v/v) DMSO, 0.2% (v/v) Triton X-100 in PBS; all from Sigma-Aldrich], containing 1.5% (v/v) goat serum, for a minimum of 2 hours at room temperature. Incubation with primary antibodies was performed

overnight at 4 °C, followed by 5 washes of 30 min with PBDX at room temperature. Slides were incubated with the secondary antibody for a minimum of 2 hours at room temperature, washed 3 times for 15 min in a solution of 0.3% (v/v) Triton X-100 in PBS (PBST) and counterstained with DAPI (1:5000; Sigma-Aldrich) for 10 min. Following 3 washes of 15 min with 1× PBS, slides were left to dry and mounted with Fluoromount-G (Invitrogen). For all antibody incubations, slides were covered with Parafilm-M (Bemis Company, Inc.) to ensure homogenous distribution of the solutions.

Primary antibodies used in this study: anti-GFP at 1:200 (chicken; GFP-1010; Aves Labs), anti-mCherry at 1:200 (chicken; when combined with rabbit anti-Aldh1a2 antibody; CPCA-mCherry; EnCor Biotechnology), anti-DsRed at 1:100 (rabbit; all other experiments; recognizing mCherry; Living Colors, 632496; Takara), anti-Aldh1a2 at 1:50 (mouse; when combined with rabbit anti-pERK or anti-Fli1 antibodies; sc-393204; Santa Cruz Biotechnology, Inc.), anti-Aldh1a2 at 1:100 (rabbit; all other experiments; GTX124302; GeneTex), anti-pERK at 1:200 (rabbit; 4370 S; Cell Signaling Technology), anti-Fli1 at 1:100 (rabbit; ab133485; Abcam), anti-Mef2 at 1:150 (rabbit; DZ01398; Boster Bio) and anti-PCNA at 1:200 (mouse; sc-56; Santa Cruz Biotechnology, Inc.), anti-Wif1 at 1:100 (rabbit; GTX16429; GeneTex), N2.261 at 1:20 (mouse; developed by H. M. Blau, obtained from the Developmental Studies Hybridoma Bank, Iowa City, IA, USA). Alexa Fluor-conjugated secondary antibodies raised in goat (Thermo Fisher Scientific) were used at 1:500. For negative control immunostainings, we used consecutive sections and followed the same protocol simultaneously but without primary antibodies. For EdU incorporation, single 20 μl intraperitoneal injections of 10 mM EdU (Thermo Fisher Scientific) per fish were performed 3 hours prior to tissue collection. The Click-iT reaction was performed after incubation with secondary antibodies using the Click-iT™ EdU Alexa Fluor™ 647 Imaging Kit (Thermo Fisher Scientific), according to manufacturer's instructions. EdU-stained samples were imaged using an Axioscan 7 microscope (Zeiss). All other immunostained samples were imaged using an LSM800 inverted confocal microscope (Zeiss). Image acquisition conditions and display intensities of the different channels were the same between groups within each experiment, for immunostained as well as for negative control sections. AFOG staining was performed as previously described[96], using the A.F.O.G. kit (BioGnost). Slides with cryosections were incubated in preheated (60 °C) Bouin's solution for 2 hours, followed by a 10 min wash in flowing tap water. Cryosections were then incubated in phosphomolybdic acid for 7 min at room temperature and washed twice with distilled water. Staining was performed with AFOG reagent for 5 min at room temperature, followed by a 3-5 min wash with flowing distilled water. Slides were dehydrated using ethanol and ROTI®Histol (both from Carl Roth), and mounted with Entellan™ (Sigma-Aldrich). Samples were imaged using a Nikon SMZ25 stereomicroscope coupled with a Nikon Digital Sight DS-Ri1 camera (Nikon) or an Axioscan 7 microscope (Zeiss).

For wholemount imaging, hearts were placed in a 0.12 mM tricaine solution (in 1× PBS) to stop the heartbeat immediately following collection, and imaged directly. Tg(cd74a:Gal4ff) hearts were imaged using a Lightsheet Z.1 light-sheet microscope (Zeiss). Tg(mhc2dab:EGFP); Tg(ptprc:DsRed) and Tg(gata4:EGFP) whole hearts were imaged using a Nikon SMZ25 stereomicroscope (Nikon).

## Tissue dissociation and cell sorting

To isolate immune cells, pools of 3 whole ventricles from Tg(mhc2dab:EGFP); Tg(ptprc:DsRed) adult zebrafish were used. Ventricles were cut into small pieces and dissociated in a 1× PBS solution containing 200 μg/ml of Liberase DL (Roche) for 20 min at 33 °C with constant agitation. The solution was gently pipetted up and down every 3 min to promote tissue disaggregation. The cell suspensions were passed through a 35 μm nylon mesh by centrifugation in a Falcon round bottom polystyrene test tube (Corning Inc.) at 300 g for 4 min at 4 °C. Samples were placed on ice and a 1:1 volume of an ice-cold 20%

FBS solution (prepared in PBS) was added. To isolate EdCs from Et(krt4:EGFP) zebrafish, pools of 2 whole ventricles (uninjured) or isolated border zones and injured tissues (post-cryoinjury) were used. To isolate cells of wild-type and cd74a; cd74b mutant samples, the injured and border zone tissues of 4 ventricles were used. For both EdCs and injured and border zone tissues, dissociation was performed using the Pierce Primary Cardiomyocyte Isolation Kit (Thermo Fisher Scientific) according to manufacturer's protocol and with a few modifications, as previously published[96]. Isolated border zone and injured tissue cells from wild types and cd74a; cd74b mutants were subsequently used for scRNA-seq. EGFP⁺ and/or DsRed⁺ cells were sorted using a FACSAria III (BD) sorter equipped with a 100 μm nozzle. Dead cells were excluded using DAPI (Sigma, Cat#D954) using 30 mW 405 nm excitation paired with a 450/50 nm band pass filter. EGFP fluorescence was measured with 50 mW 488 nm excitation paired with a 530/30 nm band pass filter. DsRed fluorescence was measured with 50 mW 561 nm excitation paired with a 586/15 nm band pass filter. As zebrafish samples are highly autofluorescent, an exclusion parameter excited by 50 mW 561 nm paired with a 610/20 nm band pass filter was used. Immune cells were sorted into a 20% FBS solution (in 1× PBS) and subsequently used for single-cell RNA-seq (scRNA-seq) analysis. EdCs were sorted into ice-cold QIAzol Lysis Reagent (Qiagen), flash-frozen in liquid nitrogen, kept at −80 °C until further processing, and used for reverse transcription quantitative PCR (RT-qPCR) analysis. The flow cytometry results were analyzed using FlowJo™ v10.8.1 software (BD Life Sciences). Example plots showing the hierarchical gating strategies can be found in Supplementary Figs. 1 and 6.

## Bulk transcriptome analysis

For bulk RNA-seq, the RNA of single ventricles was isolated using the RNeasy Micro Kit (Qiagen). The integrity of RNA and library preparation were confirmed using LabChip Gx Touch 24 (PerkinElmer). 1 μg of total RNA was used as input for VAHTS Stranded mRNA-seq Library preparation following manufacturer's protocol (Vazyme). Sequencing was performed on a NextSeq2000 instrument (Illumina) using a P3 flowcell with a 1x72bp single end setup. Raw reads were assessed for quality, adapter content and duplication rates with FastQC (http://www.bioinformatics.babraham.ac.uk/projects/fastqc). Reads with a quality drop below a mean of Q15 in a window of 5 nucleotides[97] were trimmed using Trimmomatic version 0.39. The analyses only included reads with a minimum of 15 nucleotides. Reads were trimmed, filtered, and aligned against the Ensembl zebrafish genome version danRer11 (Ensembl release 104) using STAR 2.7.9a with the parameters "--outFilterMismatchNoverLmax 0.1 --alignIntronMax 200000"[98]. Reads aligning to genes were counted with featureCounts 2.0.2 from the Subread package[99]. Only reads mapping at least partially inside exons were admitted and aggregated per gene. Reads overlapping multiple genes or aligning to multiple regions were excluded. Identification of differentially expressed genes was performed using DESeq2 version 1.30.0[100]. Significantly DEGs were classified with Benjamini-Hochberg corrected $P$ value < 0.05 and $-0.59 \leq \log_2 FC \geq +0.59$. The Ensembl annotation was enriched with UniProt data (release 12.04.2018) based on Ensembl gene identifiers (Activities at the Universal Protein Resource [UniProt]). All downstream analyses were based on the normalized gene count matrix. Volcano and MA plots were produced to highlight DEG expression. A global clustering heat map of samples was created based on the Euclidean distance of regularized log transformed gene counts. Dimension reduction analyses (PCA) were performed on regularized log transformed counts using the R packages FactoMineR[101]. DEGs were submitted to gene set overrepresentation analyses with KOBAS[102]. For the generation of heat maps of selected genes, Heatmapper (www.heatmapper.ca; University of Alberta) was used. Differential expression of selected genes was transformed to a Z-score per row and clustered using the average linkage clustering and the Euclidean distance measuring methods.

## Single-cell transcriptome analysis

Immune cells isolated from uninjured, cryoinjured (i.e., 6, 24 and 72 hpci, and 7, 14 and 30 days dpci) and 72 hours post-sham ventricles were used for scRNA-seq analysis, using the iCell8 platform. Sequencing was performed on a Nextseq2000 and raw reads were aligned against the zebrafish genome (danRer11) and counted by StarSolo[98], followed by secondary analysis in Annotated Data Format. After preprocessing, cells were analyzed via the Scanpy framework[103]. Briefly, preprocessed counts were used to calculate quality metrics in order to estimate cell quality regarding ribosomal content, mitochondrial content, number of genes, and total read count. In addition, doublets were removed by Scrublet[104]. In total, we reduced the number of cells from 1455 to 1350 high quality cells with a total of 15213 genes expressed after mitochondrial and ribosomal gene exclusion. Total numbers of cells per condition are given in Supplementary Table 1. Following QC, raw counts per cell were normalized to the median count over all cells and transformed into log space to stabilize variance. We calculated PCA and took the first 4 PCs into account for neighbors calculation[105] (15 nearest neighbors), and low-dimensional UMAP embedding[106] calculation was performed by using a minimal distance of 0.4 and a spread of 2.5. Clustering was done by Leiden with a resolution of 0.5. Final data visualization was done using a CellxGene package (doi:10.5281/zenodo.3235020). Proportion analysis of cells by cluster, timepoint, and condition was performed using the scanpro package (https://github.com/loosolab/scanpro[107]).

For single-cell transcriptomics of injured and border zone tissues of wild-type and *cd74a; cd74b* mutant samples, dead cells were removed using a LeviCell 1.0 (Levitas Bio) following tissue dissociation. The cell suspensions were counted with a Moxi cell counter and diluted according to manufacturer's protocol to obtain 10.000 single cell data points per sample. Each sample was run separately on a lane in a Chromium controller with Chromium Next GEM Single Cell 3′ Reagent Kits v3.1 (10xGenomics). scRNAseq library was prepared using standard protocol. Sample processing was performed as noted above. We removed a total of 8922 cells that did not express more than 300 genes or had a mitochondrial content greater than 70%. Furthermore, we filtered out a total of 10996 genes that were detected in less than 30 cells (< 0.01%). For this dataset, we initially reduced dimensionality of the dataset using PCA, retaining 50 principal components to calculate UMAPs. Annotation of the different clusters was performed according to the top markers of each cluster as well as the expression of known markers and by comparison with available annotated datasets[43,108,109].

## Subclustering of EdCs from scRNA-seq data

We obtained the raw count matrices of scRNA-seq from whole adult zebrafish hearts[43] (GSE159032 and GSE158919) and used the cell annotation given. We removed samples that were treated with IWR-1 and reclustered the data using Seurat v4[110]. For analysis focusing on EdCs, we subclustered these cells and integrated the datasets using Harmony[111]. Differential gene expression calculations, UMAPs, and plots of gene expression were generated using Seurat v4. Differentially expressed/top marker genes were identified using the Wilcoxon Rank Sum test, with adjusted *P* value based on Bonferroni correction.

## Analysis of scRNA-seq data of neonatal and perinatal mouse endocardial cells

Expression of *Cd74* in EdCs from P1 and P8 mice was analyzed using a scRNA-seq dataset of ventricles following left anterior descending artery ligation to induce myocardial infarction[42] (GSE153480).

## Reverse transcription quantitative polymerase chain reaction

For whole ventricle analyses, total RNA was isolated from single uninjured and cryoinjured whole ventricles per biological replicate. RNA was isolated using TRIzol Reagent (Thermo Fisher Scientific) through phenol-chloroform extraction. Total RNA was purified with the RNA Clean and Concentrator kit (Zymo Research) and a minimum of 250 ng of total RNA was reverse transcribed. For FACS-sorted EdCs, total RNA was extracted using the miRNeasy Micro Kit (Qiagen) according to manufacturer's instructions and a minimum of 200 ng of total RNA was reverse transcribed. Reverse transcription was performed using the Maxima First Strand cDNA synthesis kit (Thermo Fisher Scientific) according to manufacturer's instructions. For whole ventricles, RT-qPCR was performed in a CFX Connect Real-Time System (BioRad). For FACS-sorted EdCs, RT-qPCR was performed in a QuantStudioTM 7 pro system, 384 wells (Thermo Fisher Scientific). All reactions were performed using the DyNAmo ColorFlash SYBR Green qPCR Kit (Thermo Fisher Scientific). mRNA levels were normalized to those of *rpl13*. Fold changes were calculated using the $2^{-\Delta\Delta Ct}$ method. Ct values are included in Supplementary Tables 6, 7 and 9. Primer sequences are shown in Supplementary Table 8.

## Cyclosporine A treatments

Cyclosporine A (CsA; Sigma-Aldrich) was administered at 15 µg/g body weight through daily intraperitoneal injections from 72 hpci to 6 dpci. Briefly, a 25 mg/ml stock solution of CsA was prepared in dimethyl sulfoxide (DMSO, Sigma-Aldrich) and freshly diluted in ice-cold PBS to a 0.5 mg/ml final concentration. After removing excess water with absorbent paper, anesthetized zebrafish were weighed immediately before injection. A volume of 30 µl of a 0.5 mg/ml solution was injected per gram of zebrafish. Control animals were injected with 2% (v/v) DMSO.

## Quantifications

To quantify the density of *cd4-1*:mCherry+ cells in the deeper part of the injured tissue, the middle-most cryosection of each heart was selected and cells were counted within a 100 µm-sided square area in the center of the injured tissue (Fig. 4h). For all other quantifications, the arithmetic mean of the measurements on 2 to 3 non-consecutive cryosections per biological replicate was used. All quantifications related to pERK+ endocardium were performed using ImageJ v1.52k (Wayne Rasband, National Institutes of Health, Bethesda, MD, USA). The area of pERK+ endocardium was determined by applying a threshold to exclusively select the fluorescent area, which was then divided by the total injured area. The depth of pERK+ endocardium within the injured tissue was measured by determining the central-most (i.e., longest) segment between the pERK+ endocardial signal closest to the remote area and the one closest to the apex (Fig. 6b). The length of this segment was divided by the total length of the injury within the same axis. For all analyses, the superficial signal from the activated epicardium was excluded. EdC proliferation within the injured area was quantified by selecting random regions of interest and determining the percentage of EdU+ EdCs, as assessed by positive signal for Wif1 and DAPI. Dedifferentiating cardiomyocytes were quantified by determining the percentage of Mef2+ nuclei from N2.261+ cells, measured within a 100 µm-thick area of the border zone. Cardiomyocyte proliferation was quantified by determining the percentage of EdU+ or PCNA+ cardiomyocytes, as assessed by positive nuclear signal for Mef2+, within a 100 µm-thick area of the border zone. To determine the relative scar size, the scar area was divided by the total ventricular area. All cell density quantifications were calculated by dividing the number of cells by the injury area. For the analysis of tissue constrictions close to the injury site, all sections from each heart were assessed and a score was assigned to each heart based on the absence or presence (i.e., mild or severe) of constrictions. To analyze fibrin content in the injured tissue, we used AFOG-stained cryosections and after selecting the section from each heart with the largest fibrin content, we assigned a score based on the absence or presence (minor or major) of fibrin. To quantify myocardial wall formation, both the thickness of the muscle wall (i.e., in the middle part of the injury) and the percentage of regenerated muscle wall area close to the injured

tissue (i.e., in relation to the total area comprising the regenerated tissue and scar tissue) were measured. For this analysis, we used 2 to 3 AFOG-stained cryosections displaying the largest injury areas, and only analyzed hearts with sections showing clear boundaries between the regenerated muscle and the scar tissue.

### Statistics

For scRNA-seq cluster abundance analysis, Student's *t* test was applied for two-group comparisons, and one-way ANOVA was applied for multiple comparisons. All remaining statistics related to transcriptomic analyses can be found in the respective sections.

All other statistical analyses were performed using Prism v10.1.2 (GraphPad Software Inc., La Jolla, CA, USA). Unpaired, two-tailed Student's *t* test with Welch's correction was applied for two-group comparisons. After confirming normal distributions within groups through the Shapiro-Wilk test, one-way ANOVA and Tukey's post hoc tests were applied for multiple comparisons. Fisher's exact test was applied to test categorical variables, using the observed frequency values. Significance level was set to 0.05 for all tests.

### Illustrations

All illustrations and figures were generated using Inkscape v1.3.2.

### Reporting summary

Further information on research design is available in the Nature Portfolio Reporting Summary linked to this article.

## Data availability

All data generated in this study are provided in the Source Data file. The RNA-seq and scRNA-seq datasets generated in this study have been deposited in the National Centre for Biotechnology (NCBI) Gene Expression Omnibus database under accession codes GSE230669 and GSE232061. Source data are provided with this paper.

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

## Acknowledgements

We thank S. Howard and H.-M. Maischein for essential technical support, A. Atzberger and K. Khrievono for assistance with FACS-sorting, and K. Mattonet and S. Ramkumar for assistance in image acquisition. We also thank M. Balakrishnan, T. Molina-Villa, T.-L. Tseng, P. Goumenaki, S.-L. Lai, A. Beisaw, R. Marín-Juez, A. Bensimon-Brito, C.-C. Wu, N. Mochizuki and K. Kikuchi for discussions and critical comments on the manuscript. We further thank M. Cui for comments on the manuscript and assistance in obtaining expression data from neonatal mouse endocardial cells. Q.W. was the recipient of a scholarship from the China Scholarship Council. J.M. was the recipient of a PhD scholarship from the Studienstiftung des deutschen Volkes. Research in the Stainier Lab was supported in part by the Max Planck Society and Leducq Foundation. This work was also partly supported by a CPI Flex funds grant to J.C.-S. and D.Y.R.S.

## Author contributions

Conceptualization: J.C.-S. and D.Y.R.S. Methodology: J.C.-S. and D.Y.R.S. Investigation: J.C.-S., Q.W., P.S., J.M., S.L., S.G., M.Y., J.P. and M.L. Formal analysis: J.C.-S., Q.W., P.S., J.M., S.L., S.G., R.R., J.P., M.L. and J.P.J. Writing – Original Draft: J.C.-S. and D.Y.R.S. Writing – Review and Editing: all authors. Supervision: J.C.-S. and D.Y.R.S. Project administration: J.C.-S. and D.Y.R.S. Funding acquisition: J.C.-S. and D.Y.R.S.

## Funding

## Competing interests
The authors declare no competing interests.
