## [Peer Review File · Nature Communications]

Antigen presentation plays positive roles in the regenerative response to cardiac injury in zebrafishReviewer #1 (Remarks to the Author):

Cardeira-da-Silva et al. report that MHC presentation genes are induced by injury to the zebrafish heart and are required for normal regenerative responses. They perform profiling, use visual transgenic reporters, and generate cd74a/b mutant animals to test the idea that endocardial cells become activated by cardiac injury and serve as necessary antigen-presenting cells. This is an interesting and new idea, and the authors' data indicate that something is happening in endocardial cells that relate to antigen-presentation and interactions with CD4+ T-cells. The authors assess heart regeneration and make a conclusion that this property of endocardial cells is important for normal heart regeneration. The field would be interested in experiments assessing and ultimately supporting this model, however there are some concerns with data quality that the authors need to address for their claims to be more convincing.

Comments:

Figure 4. There are some issues with experiments in Fig. 4f-h that need to be addressed.

- Four WT samples is too few. The numbers should be at least 6-8 as in mutants.
- It should be indicated whether these 2 groups are clutchmates.
- The yellow dashed square for WT is above the injured area, while for mutant it is in the injured area. This indicates differential sampling and should be corrected.
- The high-mag images indicate a ~10-20-fold enhancement of T-cells in WT vs. mutant, whereas the quantified data indicate a 2-fold difference. Thus the image does not seem to represent the data.
- Because of the high variability of cryoinjury, the authors should show other representative samples, or all of them, in a supplement.

Figure 6c. It seems necessary to stain with pERK and an endocardial marker, so one can assess presence of endocardium and its makers of activation (pERK, Aldha2) independently. The authors need to increase their numbers to 6-8 animals, particularly in 6d, to make conclusions from these experiments.

Fig. 7c. The dedifferentiation stains and differences look convincing. However, the PCNA stain quality is below the field standards and is not convincing. Background is very high and it is not clear how the group could quantify from these samples. This experiment should be repeated or an EdU stain could be used.

Figure 7f, g. The AFOG sections are not sampled from the same area of the heart in the examples shown, and this makes it difficult to use scar percentage as a means to quantify recovery. A collagen-positive area of a certain size will be quantified as a high scar percentage in a small section toward the wall, and a low scar percentage in a large section thru the chamber (see sections in f). Tissues need to be mounted in a uniform manner, and there should be a clearly described method for quantification in the manuscript that takes this into account. These are a variable injury and assay, and I feel to be convincing the authors need to be using 10-20 hearts per timepoint, not 3-5. These data are below the field standard.

Discussion, page 11. The authors speculate "Whether the observed activation of antigen presentation genes in EdCs in the present work is highly transient or EdCs in fact differentiate to a more immune-like phenotype and down-regulate the expression of EdC markers, including Aldh1a2, is unclear."

- The Junker group has performed scRNA-seq at multiple time points during zebrafish heart regeneration, proposing that endocardial cells generate a population of transient activated fibroblasts during heart regeneration. They also co-author the current manuscript, therefore it seems interesting and important to assess the published Nat Genetics 2022 data and report whether a cluster of endocardial cell states with immune-like phenotypes is indicated in that dataset, and the dynamics of its phenotypes if so.

Other comments:

Title: The authors show evidence that endocardial cells are antigen-presenting and that MHC genes are required for heart regeneration. These may be unrelated, and thus the causal claim in their

current title does not represent the data and is too strong.

Page 3, Line 83. More accurate to refer to medaka heart as having a lower regenerative capacity. The evidence does not support an absolute term like "non-regenerative".

Page 8. The authors use the term "infiltrate" to describe the endocardium. This term is confusing as it suggests an active process by the endocardium, however there is no experimental evidence for this. They need to remove the term "infiltrating" etc. throughout the manuscript, at least as it refers to the endocardial cells.

Reviewer #2 (Remarks to the Author):

In this study, Cardeira-da-Silva and colleagues investigate the significance of adaptive immunity and antigen presentation in the process of zebrafish cardiac regeneration. Previous research has underscored the importance of a precisely regulated immune response in cardiac injury and regeneration. However, the specific immune mechanisms involved in the interplay between antigen presentation and adaptive immunity remain poorly understood. Hence, this study is both important and timely and it demonstrates that endocardial cells express genes related to antigen presentation and exhibit close associations with Cd4+ T cells. To investigate the implications of this finding, the authors created cd74a; cd74b double mutant zebrafish, thereby impairing antigen presentation. They observed that both Cd4+ T cells and activated endocardial cells were unable to infiltrate the injured area in these mutants' hearts. Consequently, the mutants displayed diminished dedifferentiation and proliferation of cardiomyocytes, indicating the necessity of these antigen presentation genes in the regenerative process. However, in contrast to this interpretation, the authors discovered that the scar area of the regenerating heart was actually reduced in the mutants at 60 days post-injury, suggesting that cd74a; cd74b might in fact suppress or delay the regenerative and/or scar resolution capacity of the heart.

This study significantly adds to our understanding of the role of MHC class II antigen presentation in cardiac regeneration. However, the connection between endocardial-specific antigen presentation, adaptive immunity, and its specific role in this process remains unclear. This, along with certain aspects of the manuscript in its present form, diminishes overall enthusiasm:

1. The authors FAC-sorted Tg(mhc2dab:EGFP)+ and/or Tg(ptprc:DsRed)+ cells from uninjured and injured ventricles to investigate "the overall immune response during zebrafish cardiac regeneration". However, the rationale for selecting these specific transgenic lines is not initially explained. This lack of clarity may confuse non-specialist readers regarding the choice of markers. To improve the presentation of the findings, it is suggested to either make the rationale for selecting these markers clearer from the beginning or rearrange the order in which the data is presented throughout the manuscript to provide a more logical flow of information.

In addition, and to provide a clearer understanding of the cell types captured by the FAC-sorting of Tg(mhc2dab:EGFP)+ and/or Tg(ptprc:DsRed)+ cells, the authors should consider including a UMAP representation that plots the distribution of mhc2dab & EGFP mRNA and ptprc & DsRed mRNA. This visualization will help establish the relationship between the identified cell clusters and the expression of these specific markers, making it easier to comprehend the cell types that are being captured by the FAC-sorting approach.

2. Related to the experiment above, the limited number of cells captured at each time point of post-injury and steady-state heart (ranging from 114 cells to 185 cells per time-point) raises concerns about the speculative and potentially biased nature of the findings derived from these initial scRNA-seq datasets. It is evident that the authors did not comprehensively capture the overall immune response in the injured heart, despite certain suggestions made in the manuscript. It is crucial to exercise caution and temper the claims made when describing the results, emphasizing the limitations imposed by the relatively small cell sample sizes. Additionally, to compare the relative abundance or frequency of different cell clusters at each timepoint, please consider representing how the distribution of cell clusters changes across the timepoints analysed.

3. The authors claim that “an unexpected link between MHC class II antigen presentation and the endocardium underlies zebrafish adult heart regeneration”. However, the observed phenotypes in the cd74a; cd74b double mutants cannot be definitively attributed to the expression of these genes by endocardial cells, given that both scRNA-seq data and transgenic analyses show cd74a and cd74b genes being expressed by various APCs, including monocytes, macrophages, lymphoid cells and EdCs. Consequently, the assertions made regarding the significance of endocardial antigen presentation and its relationship with Cd4+ T cells in cardiac regeneration lack substantial support from the data presented in the current manuscript. For this reason, the authors are advised to either moderate these claims or furnish additional data that demonstrates endocardial-specific loss of function of cd74a/b to substantiate their statements.

4. Page 6: The sentence “we found that while cd74a and cd74b expression is up-regulated in the zebrafish heart following cryoinjury, this up-regulation does not occur to the same extent in the cryoinjured medaka heart (Supplementary Fig. 2c,d)” lacks clarity regarding the authors' intended point – please clarify what the comparison suggests for non-specialist readers.

5. Page 6: “Taking advantage of the scRNA-Seq data of whole regenerating ventricles, we verified that the expression of cd4-1 is highly specific to T cells and not other cell types, including other immune cells” – In line with the same analysis, it would be informative to investigate how the expression of cd74a, cd74b, and mhc2 is distributed across endothelial cell clusters. This analysis can help determine whether all endothelial cells express antigen-presenting genes or if it is limited to a specific subtype. It is crucial to clarify this aspect and understand whether the ability to act as antigen-presenting cells (APCs) is exclusive to endocardial cells or if it is a general characteristic of endothelial cells as a whole.

6. Figure 4: The authors claim that “cd74a; cd74b double mutants exhibit reduced Cd4+ T cell infiltration of the injured tissue”. Have the authors considered the possibility of a general reduction in T cell numbers in the mutants compared to wild-type (WT) individuals? To address this concern, it would be informative to compare the levels of Cd4+ T cells between uninjured hearts of WT and mutant zebrafish. This would help determine whether the observed reduction in Cd4+ T cell infiltration is specific to the injured tissue or if it reflects an overall difference in T cell numbers between WT and cd74a/b mutants.

7. Page 7: “We found that the majority of differentially expressed genes (DEGs) were down-regulated in cd74a; cd74b double mutant ventricles compared with wild type”. It is not clear what the authors' intended point is with this sentence – please clarify what this suggests for non-specialist readers.

8. Page 8: “The cytokine-cytokine receptor interaction pathway, which may be related with various immune functions, was also down-regulated in cd74 mutants.” Which cytokine-cytokine receptor interaction pathway? Please be more specific.

9. Page 8: “MAPK signaling has previously been shown to be essential for zebrafish cardiac regeneration and to be associated with the activated endocardium in the cardiac resection model. Here, we used phospho-ERK (pERK) as a readout of MAPK signaling and confirmed its presence in Aldh1a2+ (i.e., activated) EdCs at 7 dpci (Fig. 6b).” The information of why Aldh1a2 is being used as a marker for activated EdCs should be presented earlier in the manuscript, as this would help readers understand the reasoning behind this choice.

10. Figure 6h: Providing a more detailed annotation of the identified populations in Figure 6h, along with the markers used for these annotations, would greatly enhance the informative value of the data. It would be particularly beneficial to show the expression of cd74a and cd74b within the represented clusters, as it would complement the qPCR data presented in Supplementary Table 5a. This comprehensive representation would be particularly valuable for the APCs clusters, as it would reveal the variation in cd74a and cd74b expression levels across these cell types. This is especially important considering that the mutants generated and analysed are not specific to a particular cell type.

Related to this point, in Figure 6 i,j the authors state an observed “overall reduction in the stress

response of cd74a; cd74b double mutant endocardial cells in both clusters 13 (i) and 14 (j).” - What distinguishes these endocardial cells/clusters, and could that be informative for the mutant phenotypes observed?

11. Page 8: The sentence “No striking differences were identified between wild-type and double mutant samples regarding the relative proportions of the various cell clusters (Supplementary Fig. 5c).” doesn’t seem to be substantiated by the data presented – In fact, the pie-chart representation suggests that there are higher numbers of coronary endothelial cells and cardiomyocytes in the mutant heart? Additionally, there are two cell types in the pie-chart that lack annotation (adjacent to the coronary ECs). One of these cell types, which is light pink in colour, seems to be reduced in proportion in the mutant heart. Can you please clarify these points and disparities?

12. Page 9: “We found that, compared with wild type, cd74a; cd74b double mutants exhibit a significantly reduced number of N2.261+ cardiomyocytes, indicative of decreased cardiomyocyte dedifferentiation at 120 hpci (Fig. 7a,b).” - Please analyse the scRNA-seq data to determine if there is a cluster representing a more “immature/dedifferentiation” cardiomyocyte (CM) population that might be cycling/proliferating less as compared to WT? Additionally, examine the relative proportion of this cluster within the overall cardiomyocyte cluster and assess if it correlates with the observations presented in Fig 7. This analysis could provide insights into why the pie chart represented in Supplementary Figure 5c indicates a higher presence of CMs in the mutant heart, contrary to what Fig 7 is suggesting.

13. Page 11: “Whether the observed activation of antigen presentation genes in EdCs in the present work is highly transient or EdCs in fact differentiate to a more immune-like phenotype and down-regulate the expression of EdC markers, including Aldh1a2, is unclear.” – Conducting a trajectory analysis of APCs expressing Aldh1a2 and MHC class II in both wild-type (WT) and cd74a/b mutant samples could potentially offer valuable insights into addressing this unanswered question.

14. Figure 4: please annotate in the schematics where the primers for identifying cd74a and cd74b mutants are localised.

15. Figure 7C: should read PCNA and not N2.261.

16. Figure 7F,G: please show AFOG serial sections that can help visualising and quantify the scar across the all heart.

Reviewer #3 (Remarks to the Author):

The authors studied the involvement of MHCII-based antigen-presentation during heart regeneration in the zebrafish upon cryoinjury. They provide a substantial amount of data and models to explore their hypothesis. Particularly, they compared uninjured zebrafish heart with injured ones at many different timepoints, which importantly highlight the temporal response of the immune system upon cardiac injury. The authors have found that activated endocardial cells and immune cells express MHCII, and therefore analyse further interactions with CD4+ T cells. Both activation of endocardial cells and CD4+ T cells occurs at relatively late timepoints, as compared to the presence of innate cells post-injury, notably of classical macrophages. The authors inactivate MHCII via CD74KO to show subsequent impairment of immune response and of heart regeneration. Overall, the authors hypothesis is interesting and novel, the paper is well written and the figures are clear.

Below are a few suggestions that could improve the manuscript.

- The results in Fig. 1 / S1 do not highlight the presence of dendritic cells, which are main APCs and also present in the heart / regenerating heart. Could the authors explain why? Also the neutrophils amount is higher in uninjured tissue. Is the total number of neutrophils higher or

simply the proportion? Could the authors provide the cell counts in addition to the percentages?

- At line 135, among the two clusters of lymphoid cells, cluster 7/APC is characterized by *ighv1-4* which likely refers to B cell lineage, rather than of T cells.

- At line 171 and in Figure 2c, the authors suggest that the level of expression of MHCII in the EdCs is first decrease after injury and then increases at 14dpci. Is this variation in MHCII expression directly linked to the total number of EdCs, which are expected to die following cryoinjury and proliferate subsequently, rather than an cellular increase of MHCII levels? It would be good to check the expression of MHCII in regard to the total number of EdCs.

- At line 178, the authors highlighted that not all macrophages seems to express antigen-presentation genes. Can the authors provide statistical comparison of MHCII, CD74a and CD74b expression between the different clusters in Fig. 1d? In Fig. 1d, are the data pooled from the different timepoints or are they from a specific timepoint?

- At line 178, it is unclear why the authors state that these observation reinforce the role of EdCs as major player for antigen-presentation during this process. CD74a and b expression in the qPCR analysis of the while injured ventricles show an increase after day 4, conserved up to day 7, and still present at day 14. In contrast, presentation of MHCII by endocardial cells is peaking at day 14 only.

- In Fig S1e can the authors zoom in the T cells clusters, to show whether the blue and red are overlapping populations? If not, what is the subset of "blue" T cells?

- In Fig 2b, some background are high on the microscopy images, and so it is hard to understand how the authors have placed the arrows (some on very bright or very dim sites). Did the authors have secondary antibody only controls to determine what is the specific stainings? Could the authors maybe add these control images in supplementary. Same for Fig. 6b (pERK, left top arrow?).

- In Fig 3, the authors hypothesize that the interaction of MHCII on endocardial cells is responsible for the interaction with CD4+ T cells. However, activated endocardial cells might express homing markers, like selectins and integrins that favors extravasation of the CD4+ T cells in the injured tissue. Did the authors confirmed in the double CD74 KO that the interactions between activated endocardial cells and CD4+ T cells is abolished.

- At line 214, 72h post-injury is not a timepoints with high level of MHCII presentation based on the graphs in Fig. 2-3, in which the levels are significantly lower than at day 7. In Fig. 4d, the levels between 72hpci and 7dpci are the same. Are they normalised at 1? If so, please indicate it clearly, and it would be better to not put them side by side on the same graph, as direct comparison between wt 72h and wt d7 is not possible. Same for Fig. 4e.

- Could the authors show comparison of the expression of MHCII in wild-type and mutant CD74 KO fish? Would have it been possible to do a KO for MHCII? Maybe the authors could justify their choice of CD74 in the discussion.

- At line 376:, It seems that the reference 80 focus on the role of cytokines in differentiating naive CD4+ T cells rather than detailing how macrophage activation rely on CD4+ T cells. Commonly, the reverse is known though, that CD4+ T cells activation and differentiation rely on macrophages and APCs. In the presented model, the classical macrophages are also activated and present before the infiltration of CD4+ T cells, so the hypothesis that classical macrophages might be suppressed by impaired CD4+ T cells (line 378) might need more justification.

- It seems that the ight gene specific for B cell lineage, but not for CD4+ T cells. Can the authors clarify their interest in this gene?

- Please define the dashed line in the legend of Fig 4. At line 222, is the "injured tissue" intended to be the "center of the injured tissue" as in the figure?

- Could the authors provide a statistical analysis in Fig. 6i-j, since they claim reduction in stress genes?
- In Fig. 8, I believe that the macrophage should be placed in the injured tissue rather than in the blood stream?
- It would be great to define the abbreviation IU, NTR, UAS in the text and figures (also supplementary) to facilitate the reading.
- The statistical analysis run in the manuscript and figures are not only t-test with Welch corrections as indicated in the materials and methods.
- in Fig. 7c replace N2.261 by PCNA.

RESPONSES TO REVIEWERS

Reviewer #1 (Remarks to the Author):

Cardeira-da-Silva et al. report that MHC presentation genes are induced by injury to the zebrafish heart and are required for normal regenerative responses. They perform profiling, use visual transgenic reporters, and generate cd74a/b mutant animals to test the idea that endocardial cells become activated by cardiac injury and serve as necessary antigen-presenting cells. This is an interesting and new idea, and the authors' data indicate that something is happening in endocardial cells that relate to antigen-presentation and interactions with CD4+ T-cells. The authors assess heart regeneration and make a conclusion that this property of endocardial cells is important for normal heart regeneration. The field would be interested in experiments assessing and ultimately supporting this model, however there are some concerns with data quality that the authors need to address for their claims to be more convincing.

We thank the reviewer for their supportive and constructive comments.

Comments:

1) *Figure 4. There are some issues with experiments in Fig. 4f-h that need to be addressed. - Four WT samples is too few. The numbers should be at least 6-8 as in mutants.*

We have addressed this point and the numbers are now 7-8 for each genotype.

2) *It should be indicated whether these 2 groups are clutchmates.*

It is rather challenging to obtain a large number of double mutants and double wild-type siblings from incrossing double heterozygous animals. Therefore, for comparison purposes, two sibling batches, one wild-type and one *cd74a; cd74b* double mutant, were crossed with *cd74a; cd74b* double heterozygous animals from the same batch. The generated wild-type and *cd74a; cd74b* double mutant progeny were used. This point has now been clarified in the materials and methods section.

3) *The yellow dashed square for WT is above the injured area, while for mutant it is in the injured area. This indicates differential sampling and should be corrected.*

We have now corrected this point.

4) *The high-mag images indicate a ~10-20-fold enhancement of T-cells in WT vs. mutant, whereas the quantified data indicate a 2-fold difference. Thus the image does not seem to represent the data.*

We have substituted the images with those of hearts more representative of the mean values and fold difference.

5) *Because of the high variability of cryoinjury, the authors should show other representative samples, or all of them, in a supplement.*

We have now included additional examples in Supplementary Fig. 11.

6) *Figure 6c. It seems necessary to stain with pERK and an endocardial marker, so one can assess presence of endocardium and its makers of activation (pERK, Aldha2) independently.*

In addition to pERK immunostaining, we have now included Wif1 immunostaining (new Fig. 6e). We have also included a new Supplementary Fig. 13c to show the co-localization of Wif1 and Aldh1a2 immunostaining in the injured tissue.

7) *The authors need to increase their numbers to 6-8 animals, particularly in 6d, to make conclusions from these experiments.*

We have now increased the numbers to 7-12 for all quantifications. Regarding the old Fig. 6d in particular, we do agree that the numbers were very low. However, combining different experiments may present a challenge as variations in immunostaining and image acquisition procedures between different experiments may affect intensity analysis and introduce artefacts. We also acknowledge that such variations can occur even within single experiments. Since this analysis did not add significantly to the cell and morphological data we now present, we have decided to remove it from the manuscript.

8) *Fig. 7c. The dedifferentiation stains and differences look convincing. However, the PCNA stain quality is below the field standards and is not convincing. Background is very high and it is not clear how the group could quantify from these samples. This experiment should be repeated or an EdU stain could be used.*

We have now performed EdU staining, as suggested, and substituted the PCNA data with the EdU data (Fig. 7c,d).

9) *Figure 7f, g. The AFOG sections are not sampled from the same area of the heart in the examples shown, and this makes it difficult to use scar percentage as a means to quantify recovery. A collagen-positive area of a certain size will be quantified as a high scar percentage in a small section toward the wall, and a low scar percentage in a large section thru the chamber (see sections in f). Tissues need to be mounted in a uniform manner, and there should be a clearly described method for quantification in the manuscript that takes this into account. These are a variable injury and assay, and I feel to be convincing the authors need to be using 10-20 hearts per timepoint, not 3-5. These data are below the field standard.*

We have now imaged sections from the entire heart, selected those displaying the largest scar (i.e., at the injury's midpoint), confirmed their comparability across hearts, and used them for quantification. We have also increased the numbers to 17-20 per genotype. In addition, we have incorporated a scoring analysis based on defects close to the injured tissue. This analysis involved assessing every section and assigning a score based on the absence or presence (mild or severe) of a defect. These data are now shown in Fig. 7f-h. To ensure a high number of animals analyzed, we opted to focus on the later time point (i.e., 60 dpci) and to exclude the 45 dpci time point.

10) *Discussion, page 11. The authors speculate "Whether the observed activation of antigen*

presentation genes in EdCs in the present work is highly transient or EdCs in fact differentiate to a more immune-like phenotype and down-regulate the expression of EdC markers, including Aldh1a2, is unclear.”

- The Junker group has performed scRNA-seq at multiple time points during zebrafish heart regeneration, proposing that endocardial cells generate a population of transient activated fibroblasts during heart regeneration. They also co-author the current manuscript, therefore it seems interesting and important to assess the published Nat Genetics 2022 data and report whether a cluster of endocardial cell states with immune-like phenotypes is indicated in that dataset, and the dynamics of its phenotypes if so.

We have now performed substantial subclustering analysis of endocardial cells from the published dataset (Hu et al. 2022). Among several identified subclusters, one exhibits considerable enrichment in the expression of antigen presentation genes, consistent with the immunostaining data on sections and qPCR data on sorted endocardial cells. This subcluster exhibits an immune-like transcriptome, at least compared with the other subclusters, and expresses higher levels of antigen presentation genes compared with uninjured hearts until at least 30 dpci. These new data are shown in a new Fig. 2 and Supplementary Fig. 8.

Other comments:

11) Title: The authors show evidence that endocardial cells are antigen-presenting and that MHC genes are required for heart regeneration. These may be unrelated, and thus the causal claim in their current title does not represent the data and is too strong.

The title has been changed to: Antigen presentation is required for zebrafish cardiac regeneration.

12) Page 3, Line 83. More accurate to refer to medaka heart as having a lower regenerative capacity. The evidence does not support an absolute term like “non-regenerative”.

Noted and done.

13) Page 8. The authors use the term “infiltrate” to describe the endocardium. This term is confusing as it suggests an active process by the endocardium, however there is no experimental evidence for this. They need to remove the term “infiltrating” etc. throughout the manuscript, at least as it refers to the endocardial cells.

Noted and done.

Reviewer #2 (Remarks to the Author):

In this study, Cardeira-da-Silva and colleagues investigate the significance of adaptive immunity and antigen presentation in the process of zebrafish cardiac regeneration. Previous research has underscored the importance of a precisely regulated immune response in cardiac injury and regeneration. However, the specific immune mechanisms involved in the interplay between antigen presentation and adaptive immunity remain poorly understood. Hence, this study is both important and timely and it demonstrates that endocardial cells express genes related to antigen presentation and exhibit close associations with Cd4+ T cells. To investigate the implications of this finding, the authors created cd74a; cd74b double mutant zebrafish, thereby impairing antigen presentation. They observed that both Cd4+ T cells and activated endocardial cells were unable to infiltrate the injured area in these mutants' hearts. Consequently, the mutants displayed diminished dedifferentiation and proliferation of cardiomyocytes, indicating the necessity of these antigen presentation genes in the regenerative process. However, in contrast to this interpretation, the authors discovered that the scar area of the regenerating heart was actually reduced in the mutants at 60 days post-injury, suggesting that cd74a; cd74b might in fact suppress or delay the regenerative and/or scar resolution capacity of the heart.

This study significantly adds to our understanding of the role of MHC class II antigen presentation in cardiac regeneration. However, the connection between endocardial-specific antigen presentation, adaptive immunity, and its specific role in this process remains unclear. This, along with certain aspects of the manuscript in its present form, diminishes overall enthusiasm:

We thank the reviewer for their supportive and constructive comments.

1a) The authors FAC-sorted Tg(mhc2dab:EGFP)+ and/or Tg(ptprc:DsRed)+ cells from uninjured and injured ventricles to investigate “the overall immune response during zebrafish cardiac regeneration”. However, the rationale for selecting these specific transgenic lines is not initially explained. This lack of clarity may confuse non-specialist readers regarding the choice of markers. To improve the presentation of the findings, it is suggested to either make the rationale for selecting these markers clearer from the beginning or rearrange the order in which the data is presented throughout the manuscript to provide a more logical flow of information.

The selection of these transgenic lines relied on their collective ability to label the majority of immune cell types. We have now clarified this point.

1b) In addition, and to provide a clearer understanding of the cell types captured by the FAC-sorting of Tg(mhc2dab:EGFP)+ and/or Tg(ptprc:DsRed)+ cells, the authors should consider including a UMAP representation that plots the distribution of mhc2dab & EGFP mRNA and ptprc & DsRed mRNA. This visualization will help establish the relationship between the identified cell clusters and the expression of these specific markers, making it easier to comprehend the cell types that are being captured by the FAC-sorting approach.

We have now included heat maps of mhc2dab and ptprc expression in Supplementary Fig. 4a. In addition, we have included similar heat maps for cd74a, cd74b and mhc2a expression

in Supplementary Fig. 4b. *EGFP* and *DsRed* transcripts were not picked up in this particular dataset. However, we have now included FACS-sorting plots in Supplementary Fig. 1, depicting the gating strategy to sort the cells used for scRNA-Seq analysis.

2a) *Related to the experiment above, the limited number of cells captured at each time point of post-injury and steady-state heart (ranging from 114 cells to 185 cells per time-point) raises concerns about the speculative and potentially biased nature of the findings derived from these initial scRNA-seq datasets. It is evident that the authors did not comprehensively capture the overall immune response in the injured heart, despite certain suggestions made in the manuscript. It is crucial to exercise caution and temper the claims made when describing the results, emphasizing the limitations imposed by the relatively small cell sample sizes.*

We fully agree with the reviewer and have modified the manuscript accordingly.

2b) *Additionally, to compare the relative abundance or frequency of different cell clusters at each timepoint, please consider representing how the distribution of cell clusters changes across the timepoints analysed.*

We have now included the temporal profile of the different clusters, as absolute cell number as well as fraction of cells, in Supplementary Fig. 4c,d, as suggested by the reviewer.

3) *The authors claim that “an unexpected link between MHC class II antigen presentation and the endocardium underlies zebrafish adult heart regeneration”. However, the observed phenotypes in the *cd74a*; *cd74b* double mutants cannot be definitively attributed to the expression of these genes by endocardial cells, given that both scRNA-seq data and transgenic analyses show *cd74a* and *cd74b* genes being expressed by various APCs, including monocytes, macrophages, lymphoid cells and EdCs. Consequently, the assertions made regarding the significance of endocardial antigen presentation and its relationship with *Cd4+* T cells in cardiac regeneration lack substantial support from the data presented in the current manuscript. For this reason, the authors are advised to either moderate these claims or furnish additional data that demonstrates endocardial-specific loss of function of *cd74a/b* to substantiate their statements.*

We agree with the reviewer and have modified the manuscript accordingly.

4) *Page 6: The sentence “we found that while *cd74a* and *cd74b* expression is up-regulated in the zebrafish heart following cryoinjury, this up-regulation does not occur to the same extent in the cryoinjured medaka heart (Supplementary Fig. 2c,d)” lacks clarity regarding the authors' intended point – please clarify what the comparison suggests for non-specialist readers.*

Noted and done.

5) *Page 6: “Taking advantage of the scRNA-Seq data of whole regenerating ventricles, we verified that the expression of *cd4-1* is highly specific to T cells and not other cell types, including other immune cells” – In line with the same analysis, it would be informative to investigate how the expression of *cd74a*, *cd74b*, and *mhc2* is distributed across endothelial cell clusters. This analysis can help determine whether all endothelial cells express antigen-*

presenting genes or if it is limited to a specific subtype. It is crucial to clarify this aspect and understand whether the ability to act as antigen-presenting cells (APCs) is exclusive to endocardial cells or if it is a general characteristic of endothelial cells as a whole

We have now performed a subclustering analysis of endocardial cells and identified one subcluster enriched in the expression of these antigen presentation genes and exhibiting an immune-like transcriptomic signature. These data are now incorporated in Fig. 2c and Supplementary Fig. 8b.

6) Figure 4: The authors claim that “*cd74a; cd74b* double mutants exhibit reduced *Cd4+* T cell infiltration of the injured tissue”. Have the authors considered the possibility of a general reduction in T cell numbers in the mutants compared to wild-type (WT) individuals? To address this concern, it would be informative to compare the levels of *Cd4+* T cells between uninjured hearts of WT and mutant zebrafish. This would help determine whether the observed reduction in *Cd4+* T cell infiltration is specific to the injured tissue or if it reflects an overall difference in T cell numbers between WT and *cd74a/b* mutants.

We did consider this possibility; however, we found that *Cd4+* T cells are mostly absent even in uninjured hearts. This information is now included in Supplementary Fig. 9b.

7) Page 7: “We found that the majority of differentially expressed genes (DEGs) were down-regulated in *cd74a; cd74b* double mutant ventricles compared with wild type”. It is not clear what the authors’ intended point is with this sentence – please clarify what this suggests for non-specialist readers.

Noted and done.

8) Page 8: “The cytokine-cytokine receptor interaction pathway, which may be related with various immune functions, was also down-regulated in *cd74* mutants.” Which cytokine-cytokine receptor interaction pathway? Please be more specific.

The “cytokine-cytokine receptor interaction pathway” in the KEGG pathway curated database includes genes like *csf1r*, *csf2rb*, *il6r*, *cxcl8a*, *cd40*, and *il11ra*.

9) Page 8: “MAPK signaling has previously been shown to be essential for zebrafish cardiac regeneration and to be associated with the activated endocardium in the cardiac resection model. Here, we used phospho-ERK (pERK) as a readout of MAPK signaling and confirmed its presence in *Aldh1a2+* (i.e., activated) EdCs at 7 dpci (Fig. 6b).” The information of why *Aldh1a2* is being used as a marker for activated EdCs should be presented earlier in the manuscript, as this would help readers understand the reasoning behind this choice.

Noted and done.

10a) Figure 6h: Providing a more detailed annotation of the identified populations in Figure 6h, along with the markers used for these annotations, would greatly enhance the informative value of the data. It would be particularly beneficial to show the expression of *cd74a* and *cd74b* within the represented clusters, as it would complement the qPCR data presented in Supplementary Table 5a. This comprehensive representation would be particularly valuable

for the APCs clusters, as it would reveal the variation in *cd74a* and *cd74b* expression levels across these cell types. This is especially important considering that the mutants generated and analysed are not specific to a particular cell type

Noted and done (see Supplementary Fig. 14b).

10b) Related to this point, in Figure 6 i,j the authors state an observed “overall reduction in the stress response of *cd74a*; *cd74b* double mutant endocardial cells in both clusters 13 (i) and 14 (j).” - What distinguishes these endocardial cells/clusters, and could that be informative for the mutant phenotypes observed?

All endocardial cells exhibit a reduction in the expression of genes involved in the stress response. We have now clarified this point.

11) Page 8: The sentence “No striking differences were identified between wild-type and double mutant samples regarding the relative proportions of the various cell clusters (Supplementary Fig. 5c).” doesn’t seem to be substantiated by the data presented – In fact, the pie-chart representation suggests that there are higher numbers of coronary endothelial cells and cardiomyocytes in the mutant heart? Additionally, there are two cell types in the pie-chart that lack annotation (adjacent to the coronary ECs). One of these cell types, which is light pink in colour, seems to be reduced in proportion in the mutant heart. Can you please clarify these points and disparities?

We have carried out additional analyses on the proportions of the different clusters between wild types and *cd74a*; *cd74b* mutants including statistical analysis. We do observe some differences in several clusters (new Supplementary Fig. 14c); however, none of them are statistically significant.

12) Page 9: “We found that, compared with wild type, *cd74a*; *cd74b* double mutants exhibit a significantly reduced number of N2.261+ cardiomyocytes, indicative of decreased cardiomyocyte dedifferentiation at 120 hpci (Fig. 7a,b).” - Please analyse the scRNA-seq data to determine if there is a cluster representing a more “immature/dedifferentiation” cardiomyocyte (CM) population that might be cycling/proliferating less as compared to WT? Additionally, examine the relative proportion of this cluster within the overall cardiomyocyte cluster and assess if it correlates with the observations presented in Fig 7. This analysis could provide insights into why the pie chart represented in Supplementary Figure 5c indicates a higher presence of CMs in the mutant heart, contrary to what Fig 7 is suggesting.

We have now subclustered the cardiomyocytes and identified 9 clusters, one of which consists of dedifferentiating cardiomyocytes based on the expression of genes such as *nppb*, *tnnt2a* and *mustn1b*. This cluster seems to be reduced in *cd74a*; *cd74b* mutants but statistical significance is not reached. We have incorporated these points in the text and Supplementary Fig. 15.

13) Page 11: “Whether the observed activation of antigen presentation genes in EdCs in the present work is highly transient or EdCs in fact differentiate to a more immune-like phenotype and down-regulate the expression of EdC markers, including *Aldh1a2*, is unclear.” – Conducting a trajectory analysis of APCs expressing *Aldh1a2* and MHC class II in both wild-

type (WT) and *cd74a/b* mutant samples could potentially offer valuable insights into addressing this unanswered question.

We have investigated this question using the published dataset (Hu et al., 2022), given the high number of cells and several time points included. We clustered endocardial and immune cells together to investigate whether we would find evidence for transition between these cell types. We do indeed find an intermediate population (i.e., cluster 8; Figure R1) that bridges the main endocardial and immune clusters on the UMAP. However, there is good evidence suggesting that this bridging cluster is an experimental artefact and does not correspond to a genuine cell fate transition: this cluster is characterized by strong expression of cardiomyocyte and mitochondrial genes, which is an indication of poor cell quality. With this analysis, we did not find good evidence that would confirm the presence or absence of an intermediate stage and thus would encourage a trajectory analysis. At this time, there is no conclusive evidence for EdC differentiation to immune cells. This point remains a hypothesis in the discussion section.

Figure R1. **A**, UMAP representation of endocardial and macrophage clusters from the whole regenerating zebrafish heart dataset (Hu et al., 2022). **B**, Subclustering analysis of endocardial cells

and macrophages reveals an intermediate bridging population (i.e., cluster 8). **C**, Dot plot showing strong expression of the cardiomyocyte gene *myl7* in cluster 8, indicating poor cell quality.

14) *Figure 4: please annotate in the schematics where the primers for identifying cd74a and cd74b mutants are localized.*

Done.

15) *Figure 7C: should read PCNA and not N2.261.*

We have now carried out EdU incorporation studies and have corrected this point.

16) *Figure 7F,G: please show AFOG serial sections that can help visualising and quantify the scar across the all heart.*

We have now imaged sections from the entire heart, selected those displaying the largest scar (i.e., at the injury's midpoint), confirmed their comparability across hearts, and used them for quantification. We have also increased the numbers to 17-20 per genotype. Serial sections from the entire heart are now included (Supplementary Fig. 17).

Reviewer #3 (Remarks to the Author):

The authors studied the involvement of MHCII-based antigen-presentation during heart regeneration in the zebrafish upon cryoinjury. They provide a substantial amount of data and models to explore their hypothesis. Particularly, they compared uninjured zebrafish heart with injured ones at many different timepoints, which importantly highlight the temporal response of the immune system upon cardiac injury. The authors have found that activated endocardial cells and immune cells express MHCII, and therefore analyse further interactions with CD4+ T cells. Both activation of endocardial cells and CD4+ T cells occurs at relatively late timepoints, as compared to the presence of innate cells post-injury, notably of classical macrophages. The authors inactivate MHCII via CD74KO to show subsequent impairment of immune response and of heart regeneration. Overall, the authors hypothesis is interesting and novel, the paper is well written and the figures are clear.

We thank the reviewer for their supportive and constructive comments.

Below are a few suggestions that could improve the manuscript.

1) *The results in Fig. 1 / S1 do not highlight the presence of dendritic cells, which are main APCs and also present in the heart / regenerating heart. Could the authors explain why?*

We have now re-analyzed the dataset and carefully annotated the various clusters according to the expression of top markers and a selection of known markers (Supplementary Fig. 3). From these new data, we have annotated cluster 3 as dendritic cells based on the expression of *spock3*, *ccl35.1* and *id2a* (Zhou et al., 2023) as well as *epd11*.

2) *Also the neutrophils amount is higher in uninjured tissue. Is the total number of neutrophils higher or simply the proportion? Could the authors provide the cell counts in addition to the percentages?*

We have now revisited the data and filtering procedures for the bioinformatic analysis. In summary, we have changed the filtering on cells and genes (i.e., fewer genes, more cells) and reset the UMAP parameters, resulting in a new UMAP and new clustering. As we show two dimensions, we opted to focus on the cell types in the clusters rather than on other parameters in the data. The resulting data now correlate with the biological expectations of immune cell recruitment (e.g., neutrophils peaking at 24 hpci). We have now included the temporal profile of the various clusters (both absolute cell numbers and proportion of cells) in Supplementary Fig. 4c,d.

3) *At line 135, among the two clusters of lymphoid cells, cluster 7/APC is characterized by *ighv1-4* which likely refers to B cell lineage, rather than of T cells.*

Noted and done.

4) *At line 171 and in Figure 2c, the authors suggest that the level of expression of MHCII in the EdCs is first decrease after injury and then increases at 14dpci. Is this variation in MHCII expression directly linked to the total number of EdCs, which are expected to die following*

cryoinjury and proliferate subsequently, rather than an cellular increase of MHCII levels? It would be good to check the expression of MHCII in regard to the total number of EdCs.

This qPCR analysis was indeed performed using cDNA from sorted endocardial cells. We have clarified this point in the text.

5) At line 178, the authors highlighted that not all macrophages seems to express antigen-presentation genes. Can the authors provide statistical comparison of MHCII, CD74a and CD74b expression between the different clusters in Fig. 1d?

We have now included this information in Supplementary Data 1.

6) In Fig. 1d, are the data pooled from the different timepoints or are they from a specific timepoint?

The data are from all pooled samples (i.e., uninjured, various time points post-cryoinjury and one 72 hpci post-sham sample). We have now clarified this point in the figure legend and main text.

7) At line 178, it is unclear why the authors state that these observation reinforce the role of EdCs as major player for antigen-presentation during this process. CD74a and b expression in the qPCR analysis of the while injured ventricles show an increase after day 4, conserved up to day 7, and still present at day 14. In contrast, presentation of MHCII by endocardial cells is peaking at day 14 only.

We agree that this statement was confusing and unnecessary, and have therefore removed it.

8) In Fig S1e can the authors zoom in the T cells clusters, to show whether the blue and red are overlapping populations? If not, what is the subset of "blue" T cells?

We have removed this UMAP plot and substituted it with a text citation of the dataset (Hu et al., 2022), in which the presence of T cells in the regenerating zebrafish heart is clearly described.

9) In Fig 2b, some background are high on the microscopy images, and so it is hard to understand how the authors have placed the arrows (some on very bright or very dim sites). Did the authors have secondary antibody only controls to determine what is the specific stainings? Could the authors maybe add these control images in supplementary. Same for Fig. 6b (pERK, left top arrow?).

We have repeated these immunostainings and corresponding secondary antibody only controls on consecutive sections. The experimental immunostainings are shown in Figs. 1b and 4a, and the respective controls in Supplementary Figs. 5c and 13a.

10) In Fig 3, the authors hypothesize that the interaction of MHCII on endocardial cells is responsible for the interaction with CD4+ T cells. However, activated endocardial cells might express homing markers, like selectins and integrins that favors extravasation of the CD4+ T

cells in the injured tissue. Did the authors confirmed in the double CD74 KO that the interactions between activated endocardial cells and CD4⁺ T cells is abolished.

We agree with the reviewer's comment that this physical proximity suggests a crosstalk between these cell types (as shown in our proposed model), but not necessarily (or only) antigen presentation. Cd4⁺ T cells are still observed in close physical proximity to endocardial cells in *cd74a*; *cd74b* mutants.

11) At line 214, 72h post-injury is not a timepoints with high level of MHCII presentation based on the graphs in Fig. 2-3, in which the levels are significantly lower than at day 7. In Fig. 4d, the levels between 72hpci and 7dpci are the same. Are they normalised at 1? If so, please indicate it clearly, and it would be better to not put them side by side on the same graph, as direct comparison between wt 72h and wt d7 is not possible. Same for Fig. 4e.

Noted and done.

12) Could the authors show comparison of the expression of MHCII in wild-type and mutant CD74 KO fish?

We have performed qPCR for *mhc2a* and did not observe any significant difference between genotypes (Fig. 4f).

13) Would have it been possible to do a KO for MHCII? Maybe the authors could justify their choice of CD74 in the discussion.

Noted and done.

14) At line 376:, It seems that the reference 80 focus on the role of cytokines in differentiating naive CD4⁺ T cells rather than detailing how macrophage activation rely on CD4⁺ T cells. Commonly, the reverse is known though, that CD4⁺ T cells activation and differentiation rely on macrophages and APCs. In the presented model, the classical macrophages are also activated and present before the infiltration of CD4⁺ T cells, so the hypothesis that classical macrophages might be suppressed by impaired CD4⁺ T cells (line 378) might need more justification.

We have restructured the text to highlight the role of macrophages as classical antigen presenting cells and that a potential crosstalk between Cd4⁺ T cells, EdCs, and/or macrophages may be required for cardiomyocyte repopulation as well as endocardial regeneration.

15) It seems that the *ight* gene specific for B cell lineage, but not for CD4⁺ T cells. Can the authors clarify their interest in this gene?

This gene is indeed more specific to B cells and we have therefore now removed it from the manuscript.

16) Please define the dashed line in the legend of Fig 4. At line 222, is the "injured tissue" intended to be the "center of the injured tissue" as in the figure?

It was indeed meant to refer to the center of the injured tissue. We have now corrected these points.

17) *Could the authors provide a statistical analysis in Fig. 6i-j, since they claim reduction in stress genes?*

We have now included a Supplementary Data (3) showing the list of differentially expressed genes, including the *P* values.

18) *In Fig. 8, I believe that the macrophage should be placed in the injured tissue rather than in the blood stream?*

Corrected accordingly.

19) *It would be great to define the abbreviation IU, NTR, UAS in the text and figures (also supplementary) to facilitate the reading.*

Done.

20) *The statistical analysis run in the manuscript and figures are not only t-test with Welch corrections as indicated in the materials and methods.*

Corrected accordingly.

21) *in Fig. 7c replace N2.261 by PCNA.*

We have now carried out EdU incorporation studies and have corrected this point.

References

- Hu, B. *et al.* Origin and function of activated fibroblast states during zebrafish heart regeneration. *Nat. Genet.* **54**, 1227–1237 (2022).
- Zhou, Q. *et al.* Cross-organ single-cell transcriptome profiling reveals macrophage and dendritic cell heterogeneity in zebrafish. *Cell Rep.* **42**, 112793 (2023).

Reviewer #1 (Remarks to the Author):

The authors have done a good job of increasing the rigor of their data to conform to standards of the field. As you'll see from my comments below, I feel they need to reconsider their conclusions from a key regeneration assay.

1. An important comment (#9) from my last review was that sampling and presentation did not allow the conclusion they made of disrupted heart regeneration in cd74 mutants. In their revision, the authors have more carefully handled and presented images of tissue sections from injured ventricles, sampling from the requested 10-20 animals. The quality of the data has improved.

However, the group reports more frequent "tissue constriction or defects" in mutant animals based on scoring of the shape of tissue in sections. I have not observed this description or that particular scoring system in this field. On inspection of the data, what are referred to as defects in mutants - - dimples in the wall - I see as clear evidence instead for increased presence of muscle. That is, the wild-types show poor regeneration, with exposed fibrin and collagen at the apex, while mutants actually appear to regenerate a muscular wall to replace and enclose the collagen. Surely if mutants are indeed responding better than wild-types by 60 days post injury, which the data say they are, the authors are at risk of a critical misinterpretation of these data. The authors' description of the phenotype from lines 334-341 is not compelling, and currently the title "... is required for zebrafish cardiac regeneration" is unsupported and the field would not be convinced.

2. For this work to contribute to the field, the authors should determine convincingly if and why wild-types show such major injuries at 60 dpci (in contrast to published work), and if and why mutants appear to have more muscle than wild-types in injury sites. For instance, are the initial infarct sizes similar with and without cd74 gene mutations, and have the authors stained specifically for muscle at late post-injury stages and scored this?

Even if muscle regeneration in mutants is grossly normal, the authors could publish the work with a more detailed assessment at multiple timepoints for the two assays that suggest reduced regenerative responses (dedifferentiation & cardiomyocyte proliferation). The EdU assays are suggestive though the percentages seem to be much lower than published for 7 dpci. Thus, it is still possible that antigen presentation is required for aspects of the cardiac regenerative response and that this would be a nice contribution, though clarity here and a more conservative article title are needed.

Minor: Lines 306-307. Best not to use the term "significant" so often to describe the data and potential differences. More useful to state in the text the quantified extent of differences, to help the reader determine what is significant.

Reviewer #2 (Remarks to the Author):

The study under consideration significantly contributes to our understanding of the role of MHC class II antigen presentation in cardiac regeneration. The authors have diligently revised the manuscript in response to previous feedback, incorporating necessary amendments to both temper and substantiate the presented claims. These revisions have resulted in a significantly enhanced manuscript. Notably, the evidence substantiating the antigen-presenting capabilities of endocardial cells and the essential role of MHC genes in heart regeneration has become more robust. The clarification of the relationship between these two factors is appreciated, providing a more precise representation of the data. The authors have addressed comments and I have no further suggestions.

Reviewer #3 (Remarks to the Author):

Dear Authors,

Thank you for your revisions on the manuscript. Many of my suggestions and concerns were answered properly. Some of the changes were difficult to see in the text as they were not highlighted on my version.

For example, I couldn't find where in the discussion the reasons why a KO on CD74 rather than on MHCII has been preferred (original comment 13). Same for other answers that states that the text has been changed accordingly. But overall, I saw changes have been made and the text has been improved.

For clarity purposes, I think it would be good to add the name of the cell types with the correspondent clusters in Supplementary Fig. 3 directly, rather than only in the temporal evolution of the population.

Finally, I agree now that the number of neutrophils matches the expected biological results of recruitment post injury in the new Fig.4 c, d. However, there is a population of granulocytes that remains much higher in the non-injured heart, and I wonder which granulocytes they are and why this is the case. Maybe the authors can provide an explanation or an hypothesis? (original comment 2).

RESPONSES TO REVIEWERS

Reviewer #1 (Remarks to the Author):

The authors have done a good job of increasing the rigor of their data to conform to standards of the field. As you'll see from my comments below, I feel they need to reconsider their conclusions from a key regeneration assay.

*1. An important comment (#9) from my last review was that sampling and presentation did not allow the conclusion they made of disrupted heart regeneration in *cd74* mutants. In their revision, the authors have more carefully handled and presented images of tissue sections from injured ventricles, sampling from the requested 10-20 animals. The quality of the data has improved.*

We thank the reviewer for their supportive and constructive comments.

However, the group reports more frequent "tissue constriction or defects" in mutant animals based on scoring of the shape of tissue in sections. I have not observed this description or that particular scoring system in this field. On inspection of the data, what are referred to as defects in mutants -- dimples in the wall – I see as clear evidence instead for increased presence of muscle. That is, the wild-types show poor regeneration, with exposed fibrin and collagen at the apex, while mutants actually appear to regenerate a muscular wall to replace and enclose the collagen. Surely if mutants are indeed responding better than wild-types by 60 days post injury, which the data say they are, the authors are at risk of a critical misinterpretation of these data. The authors' description of the phenotype from lines 334-341 is not compelling, and currently the title "... is required for zebrafish cardiac regeneration" is unsupported and the field would not be convinced.

*We thank the reviewer for these important points and have now gone back to examine all these samples more closely. We have now included a figure showing all the samples used (Supplementary Fig. 18). We measured myocardial wall thickness and relative area and found no significant differences between wild types and *cd74a*; *cd74b* mutants (Supplementary Fig. 19a,b). We also scored the samples based on the absence or presence (minor or major) of fibrin in the injury, and again observed no differences between genotypes (Supplementary Fig. 19c). These additional data and analyses, now included in the supplementary information, suggest that *cd74* mutants do not regenerate more muscle and are not more efficient at replacing the injured tissue compared with wild types. We have also reanalyzed the tissue constrictions and found that the tendency for increased frequency in *cd74* mutants is still present. We have modified the manuscript, including the title, accordingly.*

*2. For this work to contribute to the field, the authors should determine convincingly if and why wild-types show such major injuries at 60 dpci (in contrast to published work), and if and why mutants appear to have more muscle than wild-types in injury sites. For instance, are the initial infarct sizes similar with and without *cd74* gene mutations, and have the authors stained specifically for muscle at late post-injury stages and scored this?*

There is inherent variability when cryoinjuring tissues such as the adult zebrafish heart. To try and minimize variability, all cryoinjuries were performed by the same person using the

same equipment, including the cryoprobe, and they were conducted blindly on mixed genotypes. In order to examine whether wild types and *cd74* mutants respond differently to the initial injury, we analyzed immunostained samples from several experiments at 7 dpci and quantified the area of the injured tissue. While variable, we found no significant differences between genotypes (Fig. R1).

AFOG staining at 60 dpci is commonly used in the field (e.g., Schnabel et al., 2011; Lowe et al., 2019; Bise et al., 2020; Sharpe et al., 2022).

Figure R1. Quantification of the injury area in cryosections from wild-type and *cd74a; cd74b* mutant hearts at 7 dpci. Welch's *t* test; *P* value is included in the graph.

Even if muscle regeneration in mutants is grossly normal, the authors could publish the work with a more detailed assessment at multiple timepoints for the two assays that suggest reduced regenerative responses (dedifferentiation & cardiomyocyte proliferation).

These two assays were carried out at 120 hpci and 7 dpci to assess cardiomyocyte dedifferentiation and proliferation, respectively (Honkoop et al., 2021; Sharpe et al., 2022; Wei et al., 2023).

The EdU assays are suggestive though the percentages seem to be much lower than published for 7 dpci.

Percentages of EdU-labelled cardiomyocytes are of course dependent on how long the tissue was exposed to EdU and this exposure time varies across studies (Shoffner et al., 2020; Bertozzi et al., 2021).

In a preliminary experiment, we compared a single intraperitoneal injection administered 3 hours before heart collection with two injections given 24 hours and 3 hours before collection. We found that a 3-hour EdU incorporation period was sufficient to label cells in the border zone (Fig. R2). Therefore, to analyze cardiomyocyte proliferation, we opted for this protocol.

Figure R2. Cryosections from hearts of EdU-injected zebrafish following two different protocols: two intraperitoneal injections 24 and 3 hours before heart collection (left), and a single injection 3 hours before heart collection (right). Yellow dashed squares outline the magnified areas; dashed lines mark the border of the injured tissue. Scale bars: 100 μ m.

Thus, it is still possible that antigen presentation is required for aspects of the cardiac regenerative response and that this would be a nice contribution, though clarity here and a more conservative article title are needed.

Noted and done.

Minor: Lines 306-307. Best not to use the term "significant" so often to describe the data and potential differences. More useful to state in the text the quantified extent of differences, to help the reader determine what is significant.

Noted and done.

Reviewer #2 (Remarks to the Author):

The study under consideration significantly contributes to our understanding of the role of MHC class II antigen presentation in cardiac regeneration. The authors have diligently revised the manuscript in response to previous feedback, incorporating necessary amendments to both temper and substantiate the presented claims. These revisions have resulted in a significantly enhanced manuscript. Notably, the evidence substantiating the antigen-presenting capabilities of endocardial cells and the essential role of MHC genes in heart regeneration has become more robust. The clarification of the relationship between these two factors is appreciated, providing a more precise representation of the data. The authors have addressed comments and I have no further suggestions.

We thank the reviewer for their supportive comments.

Reviewer #3 (Remarks to the Author):

Dear Authors,

Thank you for your revisions on the manuscript. Many of my suggestions and concerns were answered properly. Some of the changes were difficult to see in the text as they were not highlighted on my version.

For example, I couldn't find where in the discussion the reasons why a KO on CD74 rather than on MHCII has been preferred (original comment 13). Same for other answers that states that the text has been changed accordingly. But overall, I saw changes have been made and the text has been improved.

We thank the reviewer for their supportive comments.

Regarding the original comment #13, the following sentences were incorporated in the discussion section:

“While the direct targeting of MHC class II genes is likely an effective strategy, the high number of these genes in zebrafish constitutes an additional challenge. Therefore, we decided to target Cd74, which is required for the assembly and trafficking of MHC class II molecules, and found that loss of Cd74 function indeed results in a compromised immune response.”

For clarity purposes, I think it would be good to add the name of the cell types with the correspondent clusters in Supplementary Fig. 3 directly, rather than only in the temporal evolution of the population.

Noted and done.

Finally, I agree now that the number of neutrophils matches the expected biological results of recruitment post injury in the new Fig.4 c, d. However, there is a population of granulocytes that remains much higher in the non-injured heart, and I wonder which granulocytes they are and why this is the case. Maybe the authors can provide an explanation or an hypothesis? (original comment 2).

These granulocytes exhibit mast-like cell characteristics, as indicated by their enriched *cpa5* and *mpx* expression (see Dobson et al. 2008). This hypothesis is further supported by the increased presence of these cells in the uninjured heart, in line with the notion of a resident and cardioprotective mast cell population observed in mammals (Parikh and Singh, 1997). Mast cells were described to contribute to the inflammatory response after tissue injury (Strbian et al., 2009; Cai et al., 2011), when they may undergo a phenotypic transition (Gentek and Hoeffel, 2017). Therefore, the observed higher number of these cells in the uninjured heart and their subsequent depletion after injury may be attributed to a transcriptomic shift towards a more inflammatory state (e.g., more neutrophil-like), possibly leading to their clustering with other inflammatory cells.

REFERENCES

- Bertozi, A. *et al.* Is zebrafish heart regeneration “complete”? Lineage-restricted cardiomyocytes proliferate to pre-injury numbers but some fail to differentiate in fibrotic hearts. *Dev. Biol.* **471**, 106-118 (2021).
- Bise, T., Sallin, P., Pfefferli, C., Jaźwińska, A. Multiple cryoinjuries modulate the efficiency of zebrafish heart regeneration. *Sci. Rep.* **10**, 11551 (2020).
- Cai, C. *et al.* Mast cells play a critical role in the systemic inflammatory response and end-organ injury resulting from trauma. *J. Am. Coll. Surg.* **213**, 604-615. (2011)
- Dobson, J. T. *et al.* Carboxypeptidase A5 identifies a novel mast cell lineage in the zebrafish providing new insight into mast cell fate determination. *Blood.* **112**, 2969-2972 (2008)
- Gentek, R., Hoeffel, G. The innate immune response in myocardial infarction, repair, and regeneration. In: *The Immunology of Cardiovascular Homeostasis and Pathology.* p251-272 (2017).
- Lowe, V. *et al.* Neupilin 1 mediates epicardial activation and revascularization in the regenerating zebrafish heart. *Development.* **146**, dev174482 (2019).
- Parikh, V., Singh, M. Resident cardiac mast cells and the cardioprotective effect of ischemic preconditioning in isolated rat heart. *J. Cardiovasc. Pharmacol.* **30**, 149-156.
- Schnabel, K., Wu, C.-C., Kurth, T., Weidinger, G. Regeneration of cryoinjury induced necrotic heart lesions in zebrafish is associated with epicardial activation and cardiomyocyte proliferation. *PLoS ONE.* **6**, e18503 (2011).
- Sharpe, M. *et al.* Ruvbl2 suppresses cardiomyocyte proliferation during zebrafish heart development and regeneration. *Front. Cell Dev. Biol.* **10**, 800594 (2022).
- Shoffner, A., Cigliola, V., Lee, N., Ou, J., Poss, K. D. Tp53 suppression promotes cardiomyocyte proliferation during zebrafish heart regeneration. *Cell. Rep.* **32**, 108089 (2020).
- Strbian, D. *et al.* An emerging role of mast cells in cerebral ischemia and hemorrhage. *Europe PMC.* **41**, 438-450.
- Honkoop, H. *et al.* Live imaging of adult zebrafish cardiomyocyte proliferation *ex vivo*. *Development.* **148**, dev199740 (2021).
- Wei, K.-H. *et al.* Comparative single-cell profiling reveals distinct cardiac resident macrophages essential for zebrafish heart regeneration. *eLife.* **12**, e84679. (2023)

Reviewer #1 (Remarks to the Author):

The authors have responded to the previous reviews and also more conservatively described their findings. I think this nice work will be of interest to the field. My one suggestion in preparing the final manuscript for publication is to consider replacing the term "scar resolution" with "scarring" or "scar tissue", which is what the AFOG assay measures at late stages after injury. Transient collagen resolves during regeneration, but whether there can be resolution of an established scar is not a consensus in the field and can be confusing language.

RESPONSES TO REVIEWERS

Reviewer #1 (Remarks to the Author):

The authors have responded to the previous reviews and also more conservatively described their findings. I think this nice work will be of interest to the field. My one suggestion in preparing the final manuscript for publication is to consider replacing the term "scar resolution" with "scarring" or "scar tissue", which is what the AFOG assay measures at late stages after injury. Transient collagen resolves during regeneration, but whether there can be resolution of an established scar is not a consensus in the field and can be confusing language.

We thank the reviewer for their supportive comments. We have replaced the term "scar resolution" accordingly.